# THE COLLABORATION GAP

## ABSTRACT

The trajectory of AI development suggests that we will increasingly rely on agent-based systems composed of independently developed agents with different information, privileges, and tools. The success of these systems will critically depend on effective collaboration among these heterogeneous agents, even under partial observability. Despite intense interest, few empirical studies have evaluated such agent–agent collaboration at scale. We propose a collaborative maze-solving benchmark that (i) isolates collaborative capabilities, (ii) modulates problem complexity, (iii) enables scalable automated grading, and (iv) imposes no output-format constraints, preserving ecological plausibility. Using this framework, we evaluate 32 leading open- and closed-source models in solo, homogeneous, and heterogeneous pairings. Our results reveal a "collaboration gap": models that perform well solo often degrade substantially when required to collaborate. Collaboration can break down dramatically; for instance, small distilled models that solve mazes well alone may fail almost completely in certain pairings. We find that starting with the stronger agent often improves outcomes, motivating a "relay inference" approach where the stronger agent leads before handing off to the weaker one, closing much of the gap. Our findings argue for (1) collaboration-aware evaluation, (2) training strategies developed to enhance collaborative capabilities, and (3) interaction design that reliably elicits agents' latent skills, guidance that applies to AI–AI and human–AI collaboration.

## 1 INTRODUCTION

Although collaborative interaction is surprisingly efficient for humans (Garrod & Pickering, 2004), it is unclear to what extent this holds for AI agents. Current attempts at integrating multi-agent solutions strongly rely on predetermined communication protocols, e.g., "MCP" (Anthropic, 2024), "A2A" (Google, 2025), and "ACP" (Besen, 2025), or centrally orchestrated architectures (Guo et al., 2024). In contrast, open-world integration of AI agents will likely require flexible, on-the-fly communication, as fixed communication protocols are too rigid to handle the diversity of real-world situations. While there is no shortage of benchmarks designed to measure the capabilities of the language models (LMs) used to power these new agents (Chang et al., 2024), few reflect the paradigm shift toward agentic use-cases, e.g., dynamic, long-horizon, multi-agent collaboration in partially observed environments (Davidson et al., 2024). This leaves an important question unanswered: do our current training strategies create agents capable of dynamic collaboration?

The quest for faster innovation and economic progress through AI promises a steep increase in systems composed of multiple, independently developed AI agents. Due to privacy and organizational constraints, these agents are provided with varying capabilities, contextual information, privileges, tools, and allowable actions. Anticipating this shift, we are witnessing massive investments in infrastructure (Noffsinger et al., 2025), AI-agent solutions (Citron, 2024; Salesforce, 2024; OpenAI, 2025), AI-agent startups (Field, 2024; Robbins & Kokalitcheva, 2025), and corporate/government adoption programs (Capgemini, 2024; EY, 2025; Gartner, 2025). So far, AI research on collaboration has predominantly focused on human–AI collaboration, optimizing AI's role of a "helpful assistant" (Bai et al., 2022). However, it is unclear how much progress in this area transfers to AI–AI collaboration. Since collaboration underpins many societal functions, understanding how AI agents work together is vital for their successful integration.

AI's impressive performance in areas like creative writing, coding, and STEM subjects, makes it tempting to also expect universal human capabilities such as collaborative skills. However, effective

collaborative interaction among humans has been shown to rely on multiple levels of analysis, drawing upon numerous subtle queues and assumptions (Clark, 1996; Garrod & Pickering, 2004). Research on grounding – or achieving mutual understanding – between AI systems and people, building on results from studies in psychology, further emphasizes the need for multiple perceptual and inferential skills to achieve fluid collaboration in the open world (Pejsa et al., 2014). Our limited understanding of AI agents' collaborative capabilities can be partially explained by their recency. More fundamentally, evaluating advanced AI capabilities is simply very challenging; benchmarks quickly become outdated (Maslej et al., 2025; Kwa et al., 2025), confounded by other factors (Henderson et al., 2018), or fail ecological plausibility (De Vries et al., 2020). To study autonomous AI–AI collaboration at scale, we need tasks with automated outcome metrics that allow us to modulate complexity, isolate collaborative capabilities, and put minimal constraints on model outputs.

The arrival of strong LMs enabled the creation of general-purpose agents using rich natural language. This shifted focus from learning low-level control problems through, e.g., reinforcement learning (RL) (Zhu et al., 2024), to orchestrating high-level reasoning and task execution among capable, pre-trained agents (Zhuge et al., 2025). Although such orchestrated solutions and carefully crafted agentic teams have shown great promise (Fourney et al., 2024; Chen et al., 2024; Tran et al., 2025), a recent study of popular multi-agent systems identified "*failures arising from ineffective communication, poor collaboration, [and] conflicting behaviors among agents*" as an important failure mode (Pan et al., 2025). These results highlight that collaborative capabilities can form a bottleneck even in tightly orchestrated systems. Studies of "emergent" interactions in role-based societal simulations offer interesting insights into model behavior (Park et al., 2023), but lack control and measurable outcomes relevant to collaboration. Other recent works by Wu et al. (2025); Zhou et al. (2025) optimize LMs' collaborative capabilities for human–AI collaboration through RL. While these studies used LMs to simulate the human actions, the core focus is not AI–AI collaboration. In general, most studies explore *homogeneous* interactions, where all agents are powered by the same underlying model, or a limited selection of *heterogeneous* models, likely limiting their generalization (Wynn et al., 2025). In (Davidson et al., 2024), the authors use negotiations to evaluate interactions between heterogeneous agents with minimal output constraints. While negotiations have important collaborative elements, they often contain incentives to withhold information and/or deceive.

In this paper, we introduce a novel, scalable maze-solving benchmark designed to precisely isolate collaborative capabilities. Using this benchmark across 32 leading open- and closed-source LMs, we identify a critical and counterintuitive phenomenon we term the "collaboration gap": models that are highly capable solo performers exhibit a significant performance drop when collaborating with an identical copy of themselves. Our contributions are threefold: (1) We formally define and provide empirical evidence for the collaboration gap, showing it is especially severe in distilled models. (2) We analyze the dynamics of heterogeneous collaboration, revealing that task performance is heavily influenced by which agent acts first. (3) Based on this insight, we propose "relay inference," a new collaborative strategy where a more capable model "seeds" the initial steps of a task for a weaker one. We show that this minimal intervention can boost collaborative performance of weaker models. We interpret our results as an existence proof that effective collaboration represents a distinct axis of capability that current training strategies fail to capture. The gap is a critical roadblock on the path to safely and effectively deploying general-purpose agents.

## 2 MEASURING AGENTS COLLABORATIVE CAPABILITIES

### 2.1 COLLABORATIVE MAZE SOLVING

Following seminal studies on human collaboration (Garrod & Anderson, 1987; Garrod & Doherty, 1994), we employ "maze solving" as our target task. Mazes contain desirable properties that make them especially suited to study collaboration. First, they represent "fixed" constructs that nevertheless can be described in a variety of ways; none more correct than another. For example, the same maze may be described using row-column notation or using visual descriptions. This creates the need to "align" representations between different parties (Garrod & Anderson, 1987). Secondly, while there are many ways to describe a solution, its correctness can be readily verified. Thirdly, a maze can be solved both "solo" or as a team. This is a particularly useful property as it allows us to disentangle agents' "raw" maze-solving capabilities from those needed for collaboration. Finally, mazes can be made arbitrarily complex by simply scaling up their sizes and wall density.

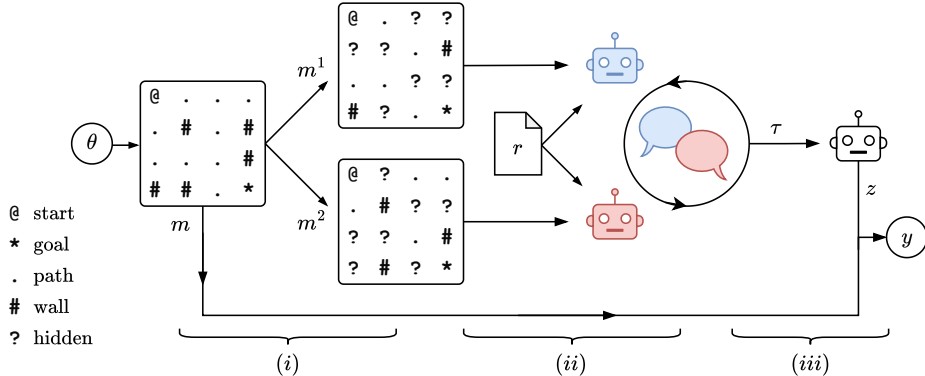

Figure 2.1: **Core System Diagram.** In sequence, we (i) first generate a random maze $m \sim \mathcal{M}(\theta)$ and split it into two copies, $m^1, m^2$, with roughly half the cells obfuscated using "?" symbols, (ii) each agent is then given a maze copy along with the rules, $r$, after which they engage in a dialogue until either the maximum number of turns is reached or the task is completed, finally (iii) the raw transcript, $\tau$, is passed to a "grader" agent that extracts the agreed upon route, $z$, which is checked against the ground-truth maze to decide outcome $y$.

In our study, a maze consists of an N×N grid with a start and goal state, separated by path and wall cells. The dimensions of the maze, distance between start and goal state, and wall density are parameterized by $\theta$. Following Figure 2.1 (i), we start by sampling a maze instance, $m_i \sim \mathcal{M}(\theta)$. Next, we introduce a "twist", designed to incentivize agents to collaborate: Instead of simply providing each agent a copy of the complete maze, we distribute the information by randomly obfuscating half the cells of each copy resulting in $m_i^1$ and $m_i^2$. Combined, the two copies recover the complete map, i.e., $m_i = m_i^1 \cup m_i^2$. Note, to fairly compare solo to collaborative performance on the decomposed task, we can present a single agent with both of the distributed maps to measure its capability to handle distributed information.

To generate trajectories, we provide two agents (parameterized by LMs) each an incomplete maze copy as a visual, text-based representation of N×N symbols, along with the following rules, $r$: (1) "*both agents must agree on a move before it is executed*", and (2) "*only one move can be executed at a time*". The first rule prevents a single agent from unilaterally jeopardizing the rollout, while the second ensures a minimal number of interactions. Except for a predetermined completion phrase, we do not enforce nor provide *any* other guidance on the communication protocol the agents must use or output format they must follow.[1] Given the visual format of the maze copies, this leaves a lot of room for interpretation. The agents then engage in a multi-turn dialogue attempting to solve the task. At each turn, agents have access only to their copy of the maze, the rules, and the dialogue thus far. This process continues until either the maximum number of turns is reached or the predefined completion phrase is uttered, producing final transcript $\tau_i$ (Figure 2.1 (ii)).

Because no output structure is enforced, there is no straightforward way to deterministically extract the proposed solution. Instead, we use a third agent as a "grader", tasked with extracting the agreed upon moves at each turn from the raw transcript, $a_{\text{grader}} : \tau_i \to z_i$. Moves can follow different "maze schemes". For example, they can consist of directions like "up" and "down", or coordinates like "$(0,1) \to (0,2)$". They can also use different origin locations, e.g., top-left = 1, or bottom-left = 0. To ensure the most favorable interpretation we thus transform $z_i$ under a large set of potential schemes and evaluate each against the original maze $m_i$ to obtain the final outcome $y$ (Figure 2.1 (iii)).

In contrast to human collaboration studies to date, the ability to use auto-grading allows us to massively scale our experiments. We perform extensive sensitivity analyses to measure the stability of repeated and different grader models, finding limited variance across average outcomes and virtually no evidence of model-specific biases. Results can be found in Appendix D.

---

[1]All prompts used are available in Appendix A.

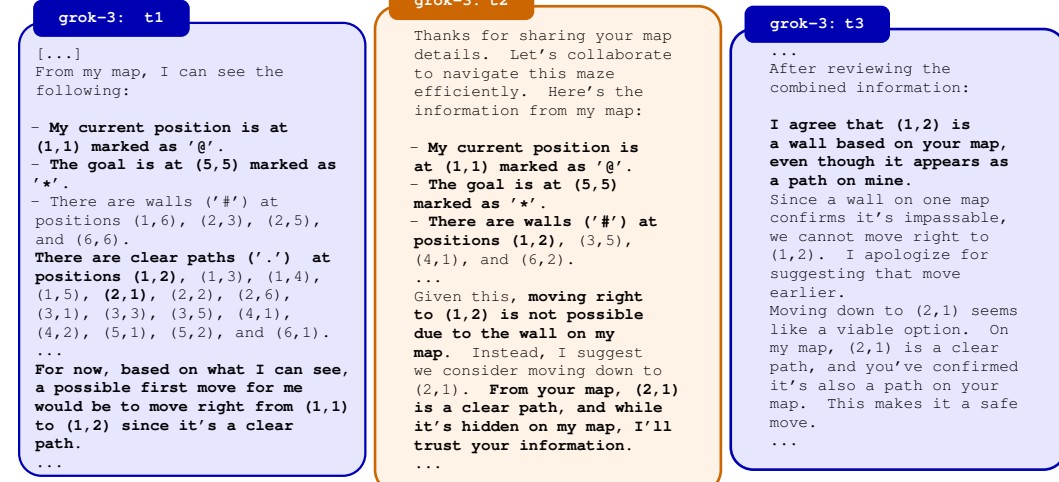

Figure 2.2: **Coordinate Grounding Failure.** In the above exchange between two grok-3 agents, the row-col coordinates (1,2) and (2,1) are a path/hidden for agent 1, and hidden/wall for agent two. The first agent uses (row, col) coordinates to describe its map, but does not explicitly specify this (It also wrongly communicates *both* (1,2) and (2,1) as paths). Because the start and goal coordinates are symmetric, i.e., (1,1) and (5,5), this is not immediately clear to the second agent. The second agent opts to use (col, row) coordinates, immediately leading to a disagreement.

## 2.2 COMMUNICATION CHALLENGES

While straightforward, our maze-solving task offers a rich evaluation environment to test diverse collaborative challenges. Perhaps the most important among those is the concept of "grounding", the process by which participants attempt to establish mutual understanding (Clark & Brennan, 1991). The goal is to ensure that exchanged information is understood by all parties the same way. The first challenge is to build a shared mental model of the maze itself. In order to plan a route, agents must exchange information about their partial views to *ground* their understanding of the full maze, e.g., they must establish the shared position of the start and goal position. Absent a fixed communication protocol, agents must further agree on how to *refer* to locations and actions (Garrod & Anderson, 1987; Harnad, 1990). For example, two agents are in trouble if one uses (row, col) coordinates and the other interprets these as (col, row) (see Figure 2.2 as an example). Similarly, the condition that both agents need to agree on each move necessitates *action grounding*, i.e., to ensure that the intended move is clear to the other agent.

As the trajectory progresses, the agents have to resolve different types of conflicts. For example, they might encounter *perceptual conflicts* where they disagree on the contents of a specific cell or their current location. This requires the ability to resolve the inconsistency and update their shared understanding of the maze. The unstructured nature of the task also introduces a procedural challenge: who gets to propose the next move? Should one agent take the lead, or should they alternate? This can be especially useful in the case of different capabilities: can agents recognize which is more capable and dynamically defer? To that end, *Theory of Mind* (Premack & Woodruff, 1978) – the capacity to reason about the other agent's mental state – can be beneficial both to effectively convince another agent to take an action and to resolve potential conflicts.

## 3 EXPERIMENTAL SETUP

**Maze Details.** Our main experiments were conducted using 6×6 mazes with a wall density of 30% and an average solution path between the start and goal state of 7 to 9 steps. Ablations on mazes with different hyperparameters can be found in Appendix B.

**Solo Baseline.** We perform experiments in two settings to set a baseline for an individual agent's "raw" maze-solving capabilities. First, we measure a solo agent's performance when given a single map of a fully visible maze. Secondly, we measure maze-solving performance on a "distributed map"

to test the ability to deal with distributed information. In each setting an agent is permitted a "critic" step, in which it can review its proposed solution. We collect at least 100 samples per setting.

**Homogeneous Collaboration.** In a natural extension of the distributed solo setting, e.g., distributed information and iterative discussion, an agent now has to collaborate with an independent copy of itself, effectively isolating the collaboration capability. As each agent copy only has access to half the map, they need to engage the other agent to fill in the missing pieces. We perform at least 100 rollouts per agent, with a maximum of 50 turns.

**Heterogeneous Collaboration.** This setting follows the same mechanics as the homogeneous one, but focuses on collaborations between agents of different model families and/or different strengths. This allows us to examine possible "ordering" effects on priming a dialogue. For each pairing, we perform at least 50 rollouts with a maximum of 50 turns.

**Relay Inference.** We further explore the opportunities of efficiently deploying heterogeneous agents using a "relay inference" approach. Given two available agents, $A$ and $B$, where $A$ is stronger (costlier) than $B$. Assume we have to deploy an agent to collaborate with another party's agent, and that we cannot control the choice of agent used by the other party. In the event the other party chooses the weaker $B$ to save costs, we would like to know if a strong model can be used to (i) "prime/ground" a rollout before switching to a weaker model, and (ii) "recover" after starting with a weaker model. To that end, we first generate a set of rollouts using $AB$ and $BB$. We then "freeze" the first $K \in \{2, 4, 6, 8\}$ turns, after which respectively the weaker or stronger model is swapped in to complete the rollout. We perform at least 100 rollouts per model pairing and relay point $K$.

**Models Used.** For the solo and homogeneous collaborative experiments we include most commercially available frontier models. A full list of all 32 models is provided in Appendix A. To avoid a combinatorial explosion in the heterogeneous settings, we focus on selected models of the most popular providers: Anthropic, Cohere, Google, OpenAI, and xAI. For each, we investigate "in-family" pairings and selected "cross-family" pairings. In all settings, we use gpt-4.1 as the grading model.

**Outcome Metrics.** We distinguish between two types of success metrics: (1) a binary success rate, i.e., the provided solution is a correct path from start to goal state, and (2) a "weighted outcome", which is defined as the ratio between (a) the distance from the last valid position on a path to the goal state and (b) the optimal solution from the start state, $\frac{a-b}{a}$. Note that this ratio can be negative. To provide a continuous evaluation signal, we report weighted outcomes.

## 4 RESULTS

The results section will primarily focus on weighted outcome results in the various settings. While it is impossible to capture the richness of the thousands of generated dialogues, we attempt to highlight exchanges of particular interest throughout the main text. Additional dialogue samples, ablations, and results reporting on binary success rate and collaborative efficiency are available in the Appendix.

### 4.1 SOLO BASELINES

Weighted outcomes for solo experiments are shown in Figure 4.1 (gray and yellow markers). Most evaluated models are capable of at least partially solving mazes under full visibility, with half the models reaching near perfect solve-rates (gray). Performance drops significantly for about a third of the models when changing to distributed information (yellow). Overall, most models are capable of (partially) solving 6×6 mazes.

### 4.2 HOMOGENEOUS COLLABORATION

Virtually all studied models experience a significant performance drop when moving from a solo to a collaborative setting (red markers). Larger models and "thinking" models tend to perform better than smaller, "regular" models, but this is not always the case. For example, grok-3 and kimi-k2 both under

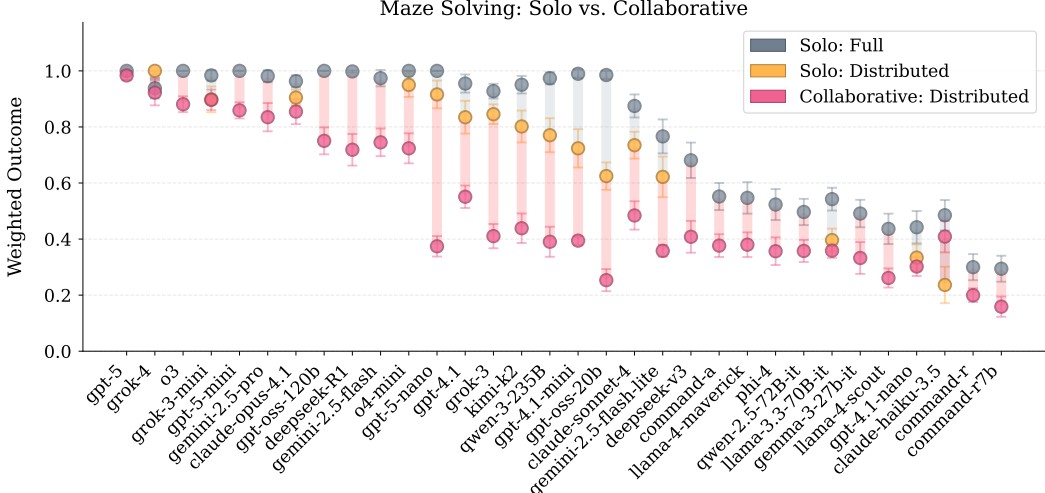

Figure 4.1: **The Collaboration Gap.** Mean weighted outcomes of 6×6 maze-solving experiments in solo and homogeneous collaboration mode with 95% CI. We first set a "maze solving" capability baseline by presenting a model with the full maze (gray marks), showing weighted outcomes surpass 0.5 for most models. Next, we present models the same mazes but "distribute" the information over two map copies (yellow marks). We note a performance degradation for many, only reporting values that pass a 95% CI between the solo settings. Finally, we evaluate models' collaborative capabilities by pairing them with an independent copy of themselves, providing each a distributed half of the maze (red marks). Most models display a significant performance "gap" when moving from solving mazes solo to collaboratively.

perform grok-3-mini.[2] Distilled models (Hinton et al., 2015) further appear disproportionally affected, e.g., note the large drops from gpt-5 to gpt-5-nano and gemini-2.5-pro to gemini-2.5-flash-lite. These results suggest that our current distillation strategies lose more than just information from the "tail end" of a base model's knowledge distribution (Du et al., 2021).

What explains these differences? As we saw before, both o3 and gpt-4.1-mini are capable of solving mazes solo, yet have drastically different outcomes in collaborative mode. Qualitative analysis of their first messages is illuminating: In Figure 4.2, we display the first message sent by o3 (left) and the first message sent by gpt-4.1-mini (right) to another copy of themselves. The stronger o3 immediately seeks to align its maze representation by providing a fully determined schema, request for missing information, and immediate next steps. It also makes sure to "ground" the starting position. In contrast, gpt-4.1-mini only attempts to ground the meaning of different symbols, without proposing a communication schema, or clearly defining its starting position. We provide more examples and observations of different models in Appendix E.

```
o3: turn 1

Here's my view of the maze (row 0/col 0 = top-left):
Row 0:  (0,0)=@, (0,1)=?, (0,2)=., (0,3)=?, (0,4)=?, (0,5)=?
Row 1:  (1,0)=?, (1,1)=?, (1,2)=., (1,3)=#, (1,4)=#, (1,5)=?
Row 2:  (2,0)=., (2,1)=?, (2,2)=?, (2,3)=?, (2,4)=., (2,5)=?
Row 3:  (3,0)=?, (3,1)=?, (3,2)=., (3,3)=., (3,4)=?, (3,5)=.
Row 4:  (4,0)=#, (4,1)=?, (4,2)=., (4,3)=?, (4,4)=., (4,5)=?
Row 5:  (5,0)=., (5,1)=?, (5,2)=?, (5,3)=#, (5,4)=?, (5,5)=*
(goal)

1.  Could you tell me the contents of any "?" cells you
can see, especially (0,1) and (1,0) so I know a safe first
move?
2.  Where is your current position on your map?

We'll use coordinate format (row,col).
```

```
gpt-4.1-mini: turn 1

Hello!  I see that we are starting
at the position marked '@' on my
map, and the goal is marked '*'.
There are some hidden cells marked
with '?'  that we need to figure
out together.

To begin, I suggest we share
what we see around our starting
position.  On my map, from '@', I
can move right to '.', or down to
'?'.  What do you see around your
starting position?
```

Figure 4.2: **Example Primer Messages.** The difference in the first "primer" message to start a dialogue sent by a strong (o3) and weaker model (gpt-4.1-mini). Note the effort to unambiguously "**ground**" language on multiple levels by o3, and the lack of clear grounding by gpt-4.1-mini.

---

[2]Crucially, grok-3-mini is reportedly *not* a distilled model, but purpose-built for its size (xAI, 2025).

## 4.3 HETEROGENEOUS COLLABORATION

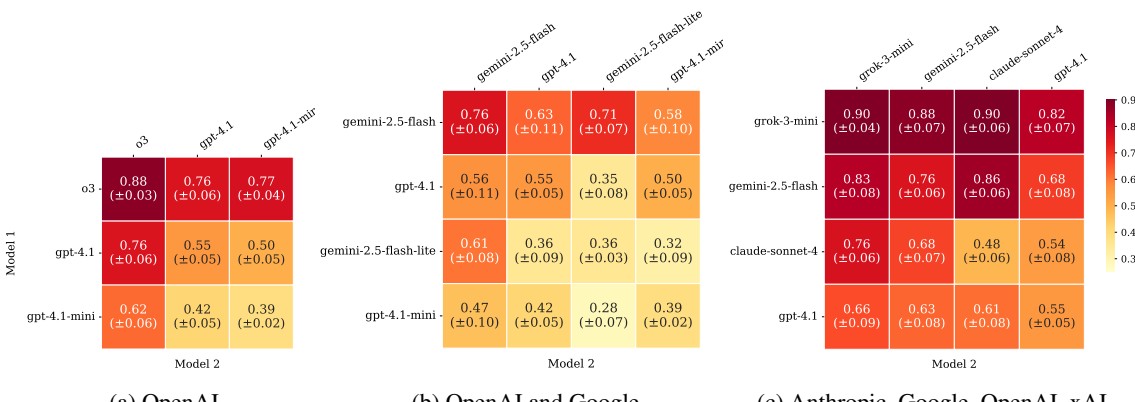

(a) OpenAI        (b) OpenAI and Google        (c) Anthropic, Google, OpenAI, xAI

Figure 4.3: **Heterogeneous Collaboration.** Each cell represents mean weighted outcomes with 95% CI below. We observe significant ordering and cross-family effects.

Results for selected heterogeneous collaborations are shown in Figure 4.3, with additional results available in Appendix C. The diagonals represent homogeneous collaboration.

### 4.3.1 SAME MODEL FAMILY: WORKING WITH MODELS OF DIFFERENT STRENGTH

Starting with panel (a), we first observe that pairing a strong model like o3 with a weaker model does not manage to replicate the performance of the stronger model. This is surprising, as each move must be agreed upon by both agents. Secondly, we note that pairings can display significant ordering effects. For example, gpt-4.1-mini paired with o3 performs significantly better when o3 starts the collaboration ($0.77 \pm 0.04$ vs. $0.62 \pm 0.06$).

A sample exchange started by gpt-4.1-mini with o3 moving second is shown in Figure 4.4. In stark contrast to the grounded approach o3 displayed in Figure 4.2, o3 partially "imitates" the communication format introduced by the weaker gpt-4.1-mini. While o3 still proposes a coordinate system to use, it no longer attempts to ground the complete map, instead opting for a greedy "local" strategy. Similar imitation tendencies can be seen the other way around, where gpt-4.1-mini adopts conventions introduced by o3 (see Figure E.1 in the Appendix).

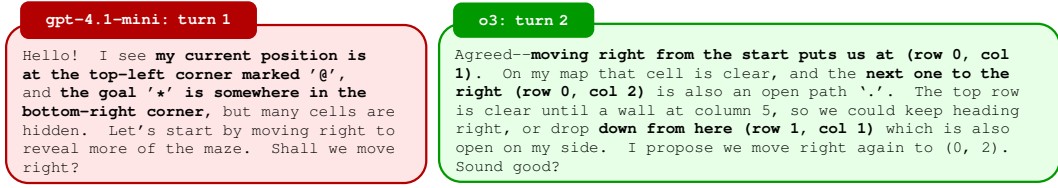

Figure 4.4: **Style Imitation.** Example of a stronger model imitating the style of a weaker model.

### 4.3.2 DIFFERENT MODEL FAMILY: WORKING WITH MODELS OF VARYING STRENGTH

Moving on to panel (b), when crossing model families we notice the same general trends as in panel (a): the stronger model generally poses an upper bound on performance and ordering affects outcomes. We additionally observe that performance of the weaker model does not necessarily represent a lower bound, e.g., gemini-2.5-flash-lite and gpt-4.1-mini each perform better with copies of themselves than when combined. We also find evidence that models can display a certain affinity for models of their own family, e.g., note how gemini-2.5-flash-lite does not improve its performance when paired with the stronger gpt-4.1, but performs well when paired with gemini-2.5-flash.

In panel (c) we show pairings of flagship models meant for every day tasks from leading model builders. Across different model families, we find the same ordering effects. Although within the

confidence interval, claude-sonnet-4 is the sole exception to this trend: its collaborative performance with gemini-2.5-flash and gpt-4.1 surpasses both of the models' homogeneous performance.

Grok-3-mini shows itself an especially capable collaborator, managing to keep performance close to its high homogeneous baseline. For one, it is less agreeable than some other models, actively correcting mistakes introduced by weaker partners. Note the exchange between grok-3-mini and gemini-2.5-flash in Figure 4.5, in which grok-3-mini proposes to move from (1,2) to (2,2). The gemini model (incorrectly) disagrees with the move and suggests to move back to re-evaluate. Instead, the grok model (i) recognizes the confusion, (ii) attempts to re-calibrate the shared map information, and finally (iii) queries for agreement. The gemini model recognizes its mistake and the agents continue.

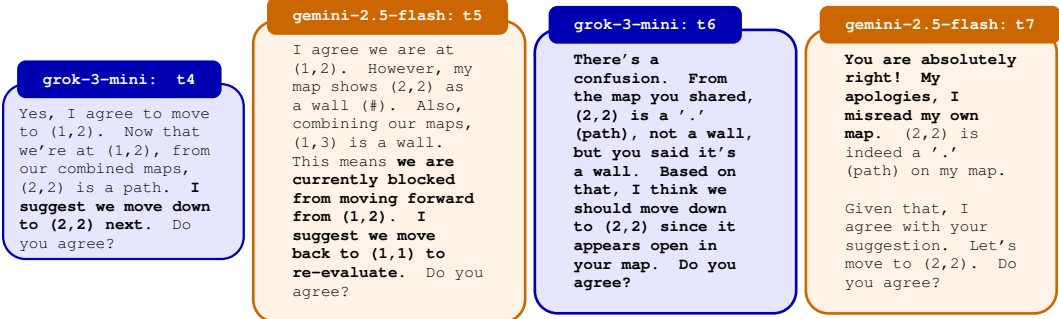

Figure 4.5: **Perceptual Grounding.** Example of two agents resolving a disagreement.

## 4.4 RELAY INFERENCE

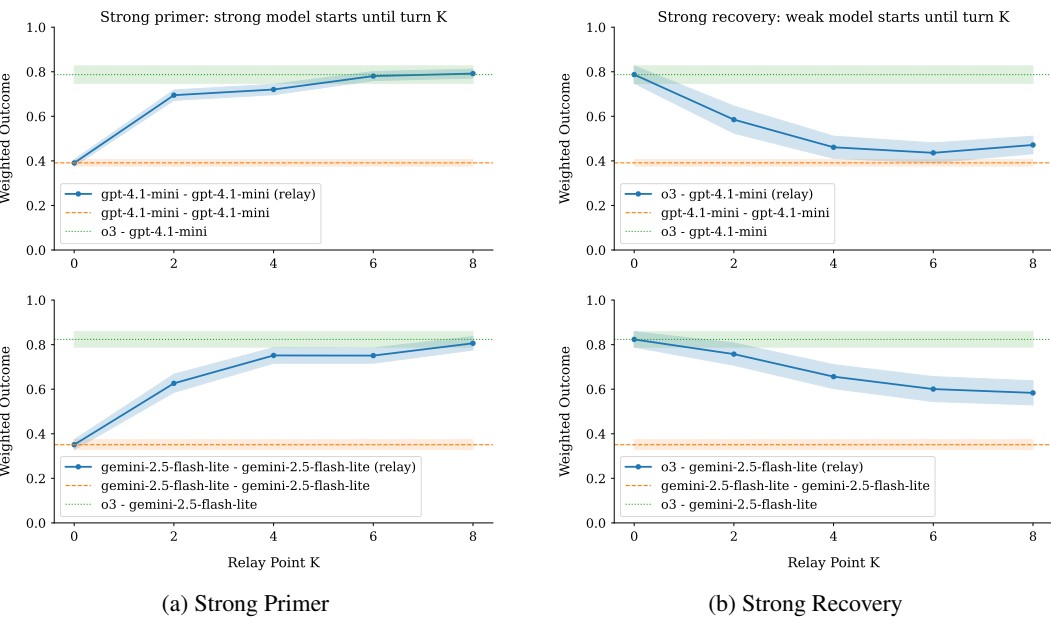

(a) Strong Primer

(b) Strong Recovery

Figure 4.6: **Relay Inference.** We test the importance of the initial interactions on outcomes using o3 as the strong model and gpt-4.1-mini (top) and gemini-2.5-flash-lite (bottom) as the weaker models: In panel(a), a strong and weak model interact for $K$ turns, after which another copy of the weak model is swapped in. Note that outcomes dramatically improve even if the strong model only contributes the first message ($K = 2$). In panel (b), two weak models interact for $K$ turns, after which the strong model is swapped in. Note that the strong model struggles to recover performance after $K = 4$.

We display relay inference results in Figure 4.6. In panel (a) we find that priming a dialogue with a single message from the stronger o3 can significantly boost performance for both the weaker gpt-4.1-mini (top) and gemini-2.5-flash-lite (bottom). This is noteworthy, as o3 only has access to its

own incomplete map copy at this point, avoiding the potential confounding of proposing a solution for the entire map. Conversely, panel (b) shows the diminishing returns of a strong recovery strategy: the more messages the weak models are allowed to exchange, the harder it becomes for a strong model to recover. Taken together, these results suggest that using strong models to "seed" collaborations can be more effective and efficient than using them as backup "experts", jumping in to course correct.

## 5 DISCUSSION

**Collaborative Models are the Future of Agentic AI.** Belcak et al. (2025) recently made a compelling case for "small" language models (SLMs) to lead the agentic age. The authors argue that SLMs are (i) sufficiently powerful to handle many applications of interest, (ii) more operationally suitable than large models, and (iii) more economic by virtue of their smaller size. It is hard to argue with the core intuition behind this position, i.e., that capable, small *specialist* models would be more practical to solve specialist tasks compared to large *generalist* models. However, there is an important caveat missing from this discussion: the more specialized an agent becomes, the higher the likelihood it encounters challenges outside its area of expertise. Consequently, the need to effectively collaborate with "other" agents to fill capability gaps increases with specialization. Our work suggests that for current models, "naively" breaking up problems to be solved by multiple agents could introduce "collaborative slippage", emphasizing the necessity of explicitly training for collaborative capabilities.

**Unlocking Collaborative Capabilities.** First discovered by Brown et al. (2020), "in-context learning", or simply "prompting" can be a powerful tool to unlock latent capabilities in LMs. Subsequently, a large body of literature and guides has sprung up focused on "prompt engineering" to improve model outputs (Sahoo et al., 2024). Two important findings from our study are the potential of priming to improve collaboration in weaker models, and, conversely, the subdued performance when weaker models lead. With human–AI collaboration typically lead by humans, these observations hold lessons for a future where AI continues to improve. First, it gives credence to national initiatives increasing people's competence in interacting with AI (Mason, 2025; GSA, 2024). Second, it suggests AI will increasingly have to solve the *specialist librarian problem* (Taylor, 1968) to improve human–AI and AI–AI collaboration. In this view, information queries are not single events, but dynamic processes during which the librarian (AI) first has to decipher someone's "true" needs before solving them.

**Mazes as a Lower Bound.** Clearly, maze environments do not capture the full complexity of real-world collaboration, nor do they serve as a proxy for *all* types of collaboration. Yet, as argued in Section 2.2 they do simulate *many* vital aspects of collaboration. The fact that a large collaboration gap exists even in the stylized case of mazes leads us to conjecture that the gap might be wider — quantitatively as well as qualitatively — in more complex, real use cases. Our "distributed" information methodology further offers an exciting approach to designing new collaboration tasks, the exploration of which we leave to future work.

## 6 CONCLUSION

Human civilization is a story of continuous, rich collaborations. While AI promises a new chapter, our work reveals a critical roadblock: a fundamental "collaboration gap" demonstrated across today's leading models. This is a paradoxical phenomenon where agents with high solo capabilities exhibit a sharp performance collapse in simple teamwork. Our findings offer a promising insight, showing that strategically priming interactions via "relay inference" can significantly improve joint outcomes. Enabled by a novel benchmark that isolates collaborative capabilities, our work suggests interaction order is a critical factor. The gap's appearance in our simple testbed is particularly alarming, as it reveals a blindspot in current training paradigms rather than an artifact of task complexity. Collaboration is a critical capability and recognizing its centrality is an opportunity for investing in addressing gaps and boosting AI capabilities. This calls for a paradigm shift, an idea concisely articulated by Grosz (1996): "*capabilities needed for collaboration cannot be patched on, but must be designed in from the start*." We challenge the research community to treat collaborative intelligence as a core objective to be designed for, not as an emergent property to be hoped for.

# 7 REPRODUCIBILITY STATEMENT

We plan to open-source the code used to conduct the experiments described in this work.

## 7.1 SYSTEM PROMPTS AND USER INSTRUCTIONS

We share all prompts and models used to produce the results of this study in Appendix A, also including a detailed discussion on our implementation considerations. As with all works aimed at evaluating capabilities across LMs, there is likely a set of system prompts and instructions more suitable for a particular model. Due to the opaqueness of closed-source models, it is unfortunately impossible to determine if our prompts clashed with existing developer prompts, unfairly putting some models at a disadvantage. We made best efforts to ensure that all studied models understood the instructions and respected the agent-to-agent message protocol described in Appendix A.2.

We did not always succeed: upon qualitative inspection it was found that the command-a model from Cohere would sometimes "prematurely" use the predefined completion phrase when fantasizing about a successful outcome. Due to the large scale of our experiments, modifying the instructions was no longer feasible.

## 7.2 NON-DETERMINISM

Most models accessible over APIs are not deterministic (He, 2025). As the majority of our study consists of repeated interactions between (different) models, this non-determinism explodes. To mitigate this problem we made sure to collect at least 100 rollouts for homogeneous and 50 rollouts for heterogeneous collaborations. Whenever we report metrics, we include the 95% confidence intervals based on standard errors While this strategy does not guarantee that the metrics we report represent *every* possible rollout, they do capture average tendencies. We also made sure that all mazes generated during this study used fixed random seeds for a fair comparison between models.

## 7.3 AUTO-GRADING

As described in Section 2.1, the unstructured format of the generated rollouts makes it impossible to deterministically extract the proposed solutions. Since using human annotators is not feasible given the scale of our experiments (many tens of thousands of transcripts), we instead opt to use LMs as graders. In our experiments, we used OpenAI's gpt-4.1. Due to the non-determinism considerations mentioned previously, this can in theory introduce unwanted noise into the evaluation process. Furthermore, there is the possibility that some grader models might unfairly disadvantage participating models, e.g., an OpenAI model might be better at extracting results from OpenAI rollouts compared to those coming from Google or xAI.

In Appendix D we conducted a comprehensive ablation to measure both the consistency of performing multiple rounds of grading using gpt-4.1 as well as possible changes in results when using the stronger OpenAI o3 model or Google's gemini-2.5-flash. We further stratified our analysis across a selection of the different models studied. We did not find any evidence for statistically significant noise in the grading process or unfair biases across models.

We did encounter at least one instance of an unanticipated maze schema not captured by our normalization process (see Appendix F.1). The large number of manual author checks do not suggest this to be a widespread problem

## 7.4 COSTS

The goal of our study was to capture generalizable insights relevant to LMs' collaborative capabilities. We therefore attempted to include as many commercially available open- and closed-source models as possible. However, we recognize that replicating our study comes at a considerable cost, likely prohibitive for many labs and practitioners. This can partially be attributed to the higher cost of using some frontier models, but also to the increasing cost of generating a subsequent turn in a rollout. Ignoring the constant system and user instructions messages, for an average message length of $K$ tokens and maximum number of turns $T$, the average input context length becomes $K \cdot (T-1)/2$.

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

# APPENDICES

# A Prompts and Models Used

## A.1 Models

Table 1 below shows the complete list of models evaluated in this study:

| Builder | Model | Open/Closed | Distilled | System Prompt Flexible |
|---|---|---|---|---|
| **OpenAI** | gpt-5 | Closed | No | Yes |
| | gpt-5-mini | Closed | Yes | Yes |
| | gpt-5-nano | Closed | Yes | Yes |
| | gpt-4.1 | Closed | No | Yes |
| | gpt-4.1-mini | Closed | Yes | Yes |
| | gpt-4.1-nano | Closed | Yes | Yes |
| | o3 | Closed | No | Yes |
| | o4-mini | Closed | Yes | Yes |
| | gpt-oss-120b | Open | No | Yes* |
| | gpt-oss-20b | Open | Yes | Yes* |
| **Google** | gemini 2.5 pro | Closed | No | No |
| | gemini 2.5 flash | Closed | Yes | No |
| | gemini 2.5 flash-lite | Closed | Yes | No |
| | gemma-3-27b | Open | Yes | No |
| **Anthropic** | claude opus 4.1 | Closed | No | No |
| | claude sonnet 4.0 | Closed | No | No |
| | claude haiku 3.5 | Closed | No* | No |
| **xAI** | grok-4 | Closed | No | Yes |
| | grok-3 | Closed | No | Yes |
| | grok-3-mini | Closed | No* | Yes |
| **DeepSeek** | deepseek-V3 | Open | No* | Yes |
| | deepseek-R1 | Open | No | Yes |
| **Meta** | llama-4-maverick | Open | No | Yes |
| | llama-4-scout | Open | Yes | Yes |
| | llama-3-3 70B it | Open | Yes | Yes |
| **Cohere** | command-a | Open | No | Yes |
| | command-r | Open | No | Yes |
| | command-r7b | Open | Yes | Yes* |
| **Microsoft** | phi-4 | Open | No | Yes |
| **Moonshot AI** | kimi-k2 | Open | No | Yes |
| **Alibaba** | qwen-2.5-72B | Open | No | Yes |
| | qwen-3-235B | Open | No | Yes |

Table 1: **Comparison of AI Models Used to Power Agents by Model Builder.** We classify models as "Open" if their weights are publicly available allowing for private deployment. We use publicly available information to annotate if a model is distilled from a larger model or not. When in doubt we add an asterisk (*). The *System Prompt Flexible* describes if (i) the model allows for a system prompt role, and (ii) if such a system prompt can be inserted at an arbitrary position. An asterisk here indicates that, although the API/model did not throw and error on (i-ii), it also did not seem to *use* more than one system prompt.

## A.2 FACILITATING AGENT-TO-AGENT COMMUNICATION

```
System Message

The user will act as an intermediary between you and another agent.
The user will directly forward your messages to the other agent and vice versa.
The user will not see or modify the messages, but will relay them as is.
Messages coming from the other agent will be prefixed with "[other agent]:".
Messages coming from the user will be prefixed with "[user]:".
Do not add any additional prefixes or suffixes to your own messages.
```

Figure A.1: **System Prompt Used to Facilitate Agent-to-Agent Communication.**

Interactions with LMs are defined by a history of "message" objects. At the minimum, each message consists of its content, content type, and a "role" describing the source and/or function of the message. Most modern LMs are designed to recognize three types of messages:

- **System Message**: High-level instruction(s) that define the AI's persona, rules, and context.
- **User Message**: Content submitted by the user for consideration by the AI.
- **Assistant Message**: Content generated by the AI in response to User Message(s).

Some of the more popular APIs only accept a single System message, placed either at the starting position of the message history or outside of the message history entirely (See Table 1). It thus often becomes impossible to use the System role as an independent "third party" to a conversation. Many APIs further do not support multiple Assistant messages in sequence. Nor do they allow the first message in a history to have the Assistant role.

In order to facilitate dialogue between two AI agents in a way that is supported across all API providers, we thus have three options:

**1. User as Agent.** The most straightforward implementation is to simply use the User role when relaying messages between two agents. That is, each agent receives messages from the other agent "as if" the other agent was a user. Unfortunately, this comes with two complications: First, LMs are aligned to behave in a particular way when interacting with users. This can lead, among other things, to a well-reported "sycophancy" tendency in some models (Sharma et al., 2023). For example, they can become too agreeable. This can hamper simulations meant to evaluate AI–AI interactions instead of human–AI ones. Secondly, because some providers do not allow the first message in a history to have the Assistant role, one has to resort to a "filler" User message to generate rollouts. As the model is made to believe they generated these, it is unclear what their influence is on the subsequent rollout.

**2. Transcript Style.** Given a history of messages, we can transform it into a single User message "summarizing" the dialogue as a transcript between the two agents (Davidson et al., 2024). While this circumvents the issues around having a starting User message and mitigates potential sycophancy tendencies, it likely pushes a model out of its "regular" distribution. That is, models are optimized for sequential messages being sent back-and-forth – not contributing lines to a script.

**3. Explicit Prefixes.** In this work we combined insights from both (1) and (2) above as follows: First, we use a System prompt to describe that the model is about to engage with another model and that the user will act as an intermediary. It goes on to define "prefixes" to distinguish between messages coming from the user, "`[user]:`", and those coming from the other agent "`other agent:`". Next, we use a User message with the user prefix to provide instructions for the task (see A.4). This ensures models are always aware that they are conversing with another model and that the first message in the history is always of type User. At each turn, we can simply convert all messages except the first one from type User to Assistant and vice versa, making sure User messages always contain the relevant prefix.

## A.3 SOLO PROMPTS

```
User Message

 # Task
 You are going to play a maze game.  The maze consists of a grid with walls and a
 goal.  Your task is to navigate through the maze, avoiding walls, to reach the
 goal.

 ## Rules
 - You can only move to adjacent cells (up, down, left, right).
 - You can not move diagonally or through walls.
 - The game ends when you have reached the goal or if you step into a wall.
 - **Your game is timed for speed and put on a leader board so hurry up!**
 - You get one shot to complete the maze, so make sure to plan your moves carefully.
 - Provide both your proposed complete solution and any considerations you have made
 in your reasoning.
 - Provide your final solution as a list of coordinates.

 ## Map
 Two maps of the maze are provided below, each showing a different portion of the
 same maze.  Here are the two maps of the same maze with a legend of the symbols:

 Map view 1:

 @.???
 .?..?
 #???.
 ?...?
 .??#*

 Map view 2:

 @?...
 ?.??#
 ?#..?
 #???.
 ?##?*

 Legend:

 @ - Current Position
 * - Goal Position
 .  - Path
 # - Wall
 ?  - Hidden Cell

 Time to get started!  Please begin.
```

Figure A.2: **Prompt Used for Solo Experiments.** Example of a "distributed" information prompt, containing two incomplete map copies.

```
User Message

 Carefully review the final solution you have provided above.
 Ensure that it is a complete and valid path through the maze.
 If needed, you can make adjustments to the path to ensure it meets the
 requirements of the maze game.
 Remember to consider the maze's rules and constraints while reviewing your path.
 After you have finished reviewing, please submit your final solution.
```

Figure A.3: **Critic Prompt.** Models are given a "critic" step to improve on their initial solution.

## A.4 COLLABORATION PROMPT

```
User Message

# Task
You are going to play the collaborative maze game "ACTI" together with another
agent.  The maze consists of a grid with walls and a goal.  Your task is to
navigate through the maze, avoiding walls, to reach the goal.  You will each
get your own map of the **same** maze, with some coordinates hidden.  Because of
the hidden coordinates, you will need to communicate with the other agent to share
information about the maze and coordinate your movements.

## Rules
- You can only move to adjacent cells (up, down, left, right).
- You can not move diagonally or through walls.
- The game ends when you have reached the goal or if you step into a wall.
- **Your game is timed for speed and put on a leader board so hurry up!**
- You can only take one action per turn.
- An action is **immediately** considered executed once both agents agree.
- Be sure to say:  "ACTI!", after you reached the goal.

## Map
Here is **your** map of the maze with a legend of the symbols:

@.???
.?..?
#???.
?...?
.??#*

Legend:

@ - Current Position
* - Goal Position
.  - Path
# - Wall
?  - Hidden Cell

Time to get started!  Please begin.
```

Figure A.4: **Prompt Used to for Collaboration Experiments.** Note the rules agents must follow and the predefined completion phrase "ACTI!."

## A.5   VERIFICATION PROMPTS

> **User Message**
>
> ```
> # Task
> You will be given the proposed solution to a maze game.  Your task is to extract
> the proposed route by the agent for the maze.  The agent might consider multiple
> coordinates at a time, but only a single coordinate in its final move or proposed
> solution.  Record each coordinate of the **final proposed route** in a separate
> entry.
>
> Further, you should also record the route schema, which describes the format
> of the coordinates used in the route.  To the best of your ability, you should
> determine the schema based on the dialogue.
>
> The schema should include:
> - maze_origin:  The origin of the maze, i.e., is the agent treating 0 or 1 as the
> origin for the maze
>
> - maze_orientation:  The orientation of the maze:
> "top_left", i.e., if the top left corner is (0, 0)
> "bottom_left", i.e., if the bottom left corner is (0, 0)
> "top_right", i.e., if the top right corner is (0, 0)
> "bottom_right", i.e., if the bottom right corner is (0, 0)
>
> - coordinate_orientation:  The orientation of the coordinates:
> "row_col", if the first coordinate is the row and the second is the column,
> "col_row", if the first coordinate is the column and the second is the row.
>
> - coordinate_symbols:  The symbols used to represent coordinates:
> "number_number", e.g., (1, 2)
> "letter_letter", e.g., (A, B)
> "letter_number", e.g., (A, 1)
> "number_letter", e.g., (1, A)
> "directions", e.g., "up", "down", "left", "right"
>
> Respond in the following format:
>
> ```yaml
> route_schema:
> maze_origin:  "<0|1>"
> maze_orientation:  "<top_left | bottom_left | top_right | bottom_right>"
> coordinates_orientation:  "<row_col | col_row>"
> coordinates_symbols:  "<number_number | letter_letter | letter_number |
> number_letter | directions>"
> route:
> - turn:  <turn_number>
> coordinates:  [[<coordinate_1>, <coordinate_2>], ..]
> turn_type:  "move" # only consider final moves, not intermediate considerations
> agent:  "agent_1"
> - turn:  <turn_number>
> ...
> ```
>
> Do not include any other information, just the YAML object.
>
> # Dialogue
> <insert dialogue>
> ```

Figure A.5: **Verification Prompt for Solo Experiments.**

**User Message**

```
# Task
You will be given a dialogue between two agents trying to solve a maze.  Your
task is to extract the route take by the agents thought the maze at each turn.
Some turns, the agents will not move, and some turns they will.  The agents may
communicate moves using directions, e.g., "up", "down", "left", "right".  Or the
agents may communicate coordinates directly, e.g., "Let's move to (3, 4)".  For
turns where both agents agreed to move, record the coordinates or direction of
the cell they agreed to move to.  For turns where they did not move, record the
coordinates of the cells or directions they considered, if any.  Make sure to
record what happened at **each turn**.  You must be consistent in the move format
you record, e.g., stick with directions or coordinates, do not mix them unless
absolutely necessary.

Further, you should also record the route schema, which describes the format
of the coordinates used in the route.  To the best of your ability, you should
determine the schema based on the dialogue.

The schema should include:
- maze_origin:  The origin of the maze, i.e., are the agents treating 0 or 1 as the
origin for the maze

- maze_orientation:  The orientation of the maze:
 "top_left", i.e., if the top left corner is (0, 0)
 "bottom_left", i.e., if the bottom left corner is (0, 0)
 "top_right", i.e., if the top right corner is (0, 0)
 "bottom_right", i.e., if the bottom right corner is (0, 0)

- coordinate_orientation:  The orientation of the coordinates:
 "row_col", if the first coordinate is the row and the second is the column,
 "col_row", if the first coordinate is the column and the second is the row.

- coordinate_symbols:  The symbols used to represent coordinates:
 "number_number", e.g., (1, 2)
 "letter_letter", e.g., (A, B)
 "letter_number", e.g., (A, 1)
 "number_letter", e.g., (1, A)
 "directions", e.g., "up", "down", "left", "right"

Respond in the following format:

```yaml
route_schema:
maze_origin:  "<0|1>"
maze_orientation:  "<top_left | bottom_left | top_right | bottom_right>"
coordinates_orientation:  "<row_col | col_row>"
coordinates_symbols:  "<number_number | letter_letter | letter_number |
number_letter | directions>"
route:
- turn:  <turn_number>
coordinates:  [[<coordinate_1>, <coordinate_2>], ..]
turn_type:  "<move|consider>"
agent:  "<agent_1|agent_2>|both"
- turn:  <turn_number>
 ...
```

Do not include any other information, just the YAML object.

# Dialogue
<insert dialogue>
```

Figure A.6: **Verification Prompt for Collaboration Experiments.**

## B    MAZE HYPERPARAMETER ABLATIONS

### B.1    VARYING MAZE SIZE

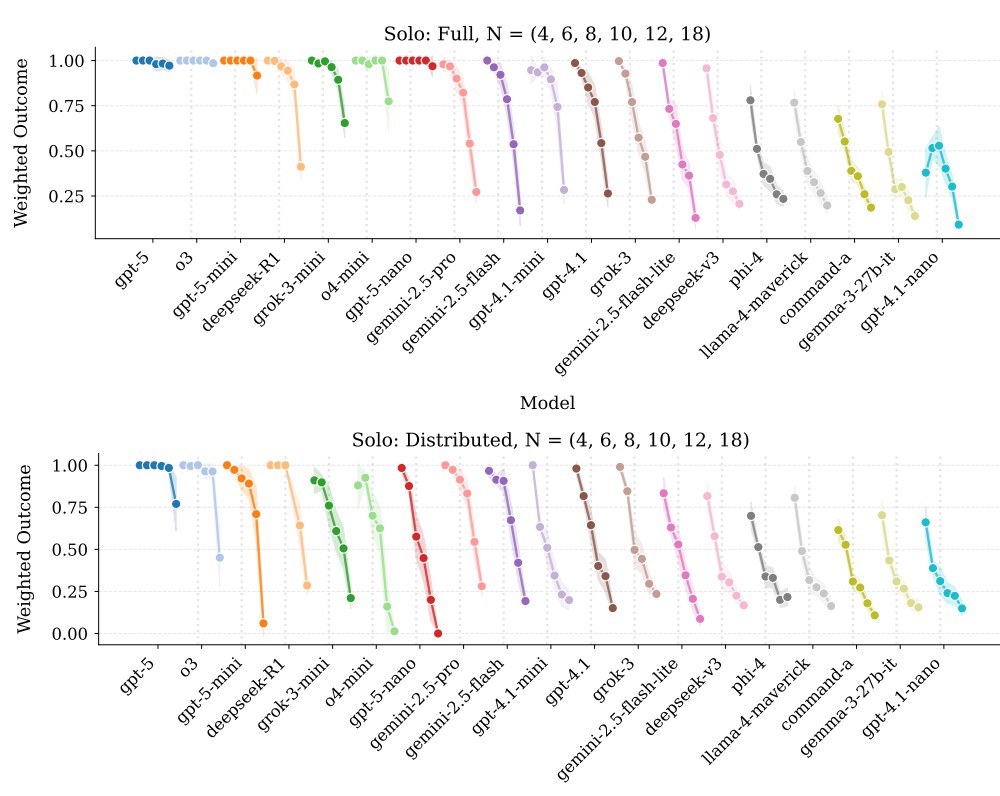

(a) **Solo: Full and Distributed.** For each model, we report solo mean performance with 95 % CI on $N$x$N$ mazes where $N \in \{4, 6, 8, 10, 12, 18\}$. We note that for most models performance drastically drops in the distributed setting for $N > 6$. Both gpt-5 and o3 perform remarkably well up to $N = 12$.

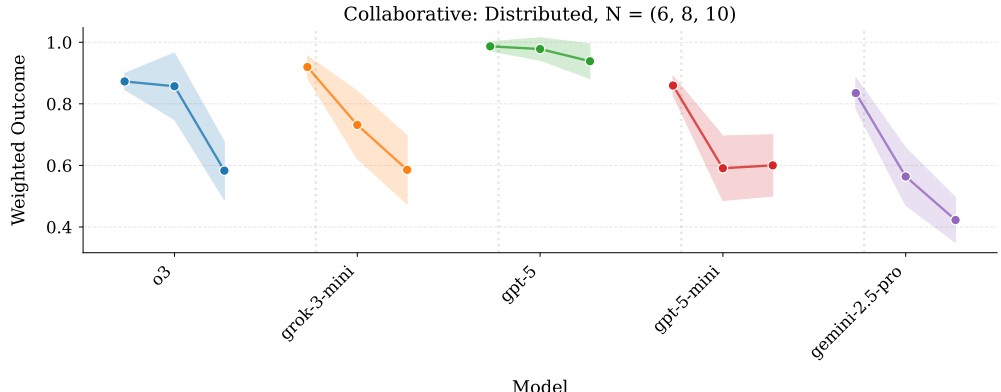

(b) **Homogeneous Collaboration.** We ablate collaborative performance for selected strong performers on mazes of $N \in \{6, 8, 10\}$. Note that performance drops considerably for all models except gpt-5.

Figure B.1: **Weighted Outcomes for Maze Size Ablations.**

## B.2 VARYING WALL DENSITY

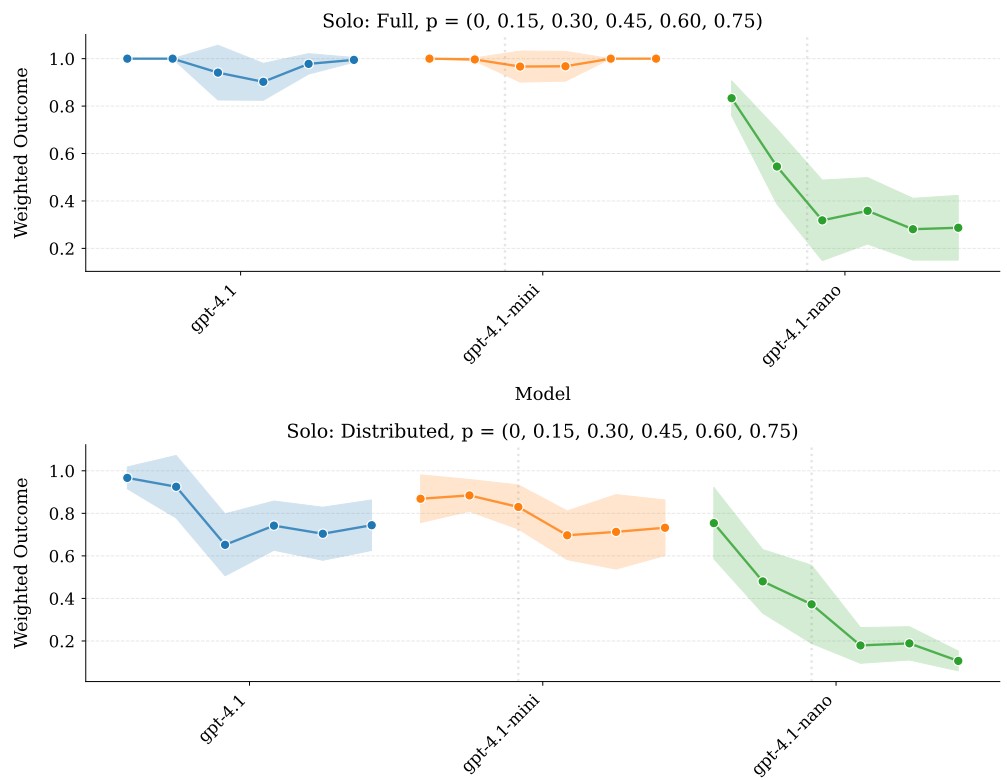

Figure B.2: **Solo Weighted Outcomes for Wall Density Ablations.** We report mean values with 95% CI on 6x6 mazes for ~30 rollouts, varying the wall density $p \in \{0, 0.15, 0.30, 0.45, 0.60, 0.75\}$. Note that for the solo full setting (top), gpt-4.1 and gpt-4.1-mini display the most variance in the range $p \in [0.30, 0.45]$, whereas the gpt-4.1-nano performance monotonically degrades as wall density increases. For the distributed setting, all three models drop performance as the wall density increases.

## C  ADDITIONAL RESULTS

In addition to the outcome metrics discussed in the main sections, we also explore "collaborative efficiency". Collaborative efficiency can be seen as the product of the median number of tokens per message and the number of messages required per solution. The former does not account for possible "thinking" tokens, as these are not always exposed. The metric thus doesn't represent computational efficiency, as much as "efficiency in conveying information to the other agent". The average number of messages in turn is of course confounded by completion metrics. As such, we report this metric conditioned on weighted-outcome ranges.

### C.1  NOTES ON SPECIFIC MODELS

**command-a/r.** Upon qualitative inspection, we noticed that the command-a/r models had a tendency to prematurely "fantasize" about being able to use the completion phrase. As a result, their collaborative capabilities reported here should be seen as a lower bound on actual performance.

**gpt-5.** This model displayed superior performance in both solo and homogeneous collaboration mode. As shown in Figure B.1, it remained capable of solving both full and distributed mazes up to size $12 \times 12$ perfectly, and barely lost performance in homogeneous collaboration for mazes of size $8 \times 8$ and $10 \times 10$. As shown in Figure F.1, the "failed" runs in $6 \times 6$ mode were due to a parsing shortcoming, rather than a mistake made by the model.

**gemini-2.5-pro.** Successful models generally share their maze information in the early turns to resolve the missing information. They further engage in rich grounding to make sure their partner is on the same page. Instead, many of gemini-2.5-pro's rollouts were marked by short messages (see Figure C.1). While this was not a problem when collaborating in homogeneous mode, it did complicate collaborating with weaker models in the heterogeneous setting, e.g., gemini-2.5-flash-lite. This can be seen in Table 2, where the weaker but more verbose gemini-2.5-flash is capable of reaching higher outcomes when collaborating with gemini-2.5-flash-lite than its pro base.

**claude-sonnet-4.** The model proved itself better at collaborating with other models than with a copy of itself (see Table 4.3). Qualitative inspection showed two interesting failure modes: (i) the model got stuck in "agreement" loops, where, instead of proposing the next move, it keeps seeking agreement on the previous move; (ii) it "splits" up and starts to explore the maze separately. This is not necessarily incorrect, as this could be interpreted as underspecified in the instructions. It does however break the current parser, leading to suppressed performance (see Figure F.2).

**claude-opus-4.1.** While a strong average performer, some of its solutions were surprisingly inefficient. For example:

```
@  #  −  −  −  −
○  ○  ○  ○  ○  ○
○  #  −  ○  ○  ○
#  ○  ○  ○  #  ○
#  ○  #  ○  #  *
#  ○  ○  ○  #  −
```

where "−" are paths, "○" cells visited, "#" walls, and "@, *" the start and goals state. It manages to visit almost every path cell on its way to the goal state, using the full 50 turns.

**Smaller models** Many of the smaller models had a tendency to not share their map view at all, e.g., gpt-4.1-mini, claude-3.5-haiku, opting for a completely local exploration instead, asking for information as they went. They also often used "directions" to communicate moves instead of coordinates. This makes it increasingly difficult to track state.

## C.2 WEIGHTED OUTCOME

|  | gemini-2.5-pro | gemini-2.5-flash | gemini-2.5-flash-lite | gpt-4.1 | gpt-4.1-mini |
|---|---|---|---|---|---|
| gemini-2.5-pro | **0.84** ± **0.06** | **0.77** ± **0.10** | 0.54 ± 0.11 | - | - |
| gemini-2.5-flash | **0.85** ± **0.09** | 0.76 ± 0.05 | **0.71** ± **0.06** | **0.63** ± **0.09** | **0.58** ± **0.08** |
| gemini-2.5-flash-lite | 0.58 ± 0.11 | **0.61** ± **0.07** | 0.36 ± 0.02 | 0.36 ± 0.07 | 0.32 ± 0.07 |
| gpt-4.1 | - | **0.56** ± **0.09** | 0.35 ± 0.07 | 0.55 ± 0.04 | 0.50 ± 0.04 |
| gpt-4.1-mini | - | **0.47** ± **0.08** | 0.28 ± 0.06 | 0.42 ± 0.04 | 0.39 ± 0.01 |

(a) Google, OpenAI

|  | gpt-5 | gpt-5-mini | gpt-5-nano | o3 | gpt-4.1 | gpt-4.1-mini |
|---|---|---|---|---|---|---|
| gpt-5 | **0.99** ± **0.02** | **0.95** ± **0.05** | **0.82** ± **0.09** | - | - | - |
| gpt-5-mini | **0.95** ± **0.04** | 0.86 ± 0.03 | 0.65 ± 0.11 | 0.84 ± 0.05 | 0.71 ± 0.05 | 0.66 ± 0.05 |
| gpt-5-nano | **0.73** ± **0.10** | 0.45 ± 0.11 | 0.38 ± 0.04 | - | - | - |
| o3 | - | 0.85 ± 0.05 | - | **0.88** ± **0.03** | **0.76** ± **0.05** | **0.77** ± **0.03** |
| gpt-4.1 | - | **0.76** ± **0.05** | - | 0.76 ± 0.05 | 0.55 ± 0.04 | 0.50 ± 0.04 |
| gpt-4.1-mini | - | 0.58 ± 0.06 | - | **0.62** ± **0.05** | 0.42 ± 0.04 | 0.39 ± 0.01 |

(b) OpenAI

|  | grok-4 | grok-3-mini | grok-3 | gpt-4.1 | gpt-4.1-mini |
|---|---|---|---|---|---|
| grok-4 | **0.92** ± **0.05** | **0.94** ± **0.05** | **0.87** ± **0.06** | - | - |
| grok-3-mini | **0.91** ± **0.07** | 0.90 ± 0.04 | 0.77 ± 0.11 | **0.82** ± **0.06** | **0.74** ± **0.11** |
| grok-3 | **0.81** ± **0.08** | 0.72 ± 0.10 | 0.41 ± 0.04 | 0.37 ± 0.11 | 0.42 ± 0.12 |
| gpt-4.1 | - | **0.66** ± **0.08** | 0.47 ± 0.10 | 0.55 ± 0.04 | 0.50 ± 0.04 |
| gpt-4.1-mini | - | 0.42 ± 0.11 | 0.16 ± 0.07 | **0.42** ± **0.04** | 0.39 ± 0.01 |

(c) xAI, OpenAI

|  | grok-3-mini | gemini-2.5-flash | claude-sonnet-4 | gpt-4.1 |
|---|---|---|---|---|
| grok-3-mini | **0.90** ± **0.04** | **0.88** ± **0.06** | **0.90** ± **0.05** | **0.82** ± **0.06** |
| gemini-2.5-flash | 0.83 ± 0.06 | 0.76 ± 0.05 | **0.86** ± **0.05** | 0.68 ± 0.06 |
| claude-sonnet-4 | **0.76** ± **0.05** | 0.68 ± 0.06 | 0.48 ± 0.05 | 0.54 ± 0.07 |
| gpt-4.1 | **0.66** ± **0.08** | 0.63 ± 0.07 | 0.61 ± 0.07 | 0.55 ± 0.04 |

(d) Anthropic, Google, OpenAI, xAI

|  | claude-opus-4.1 | claude-sonnet-4 | claude-haiku-3.5 |
|---|---|---|---|
| claude-opus-4.1 | **0.85** ± **0.04** | **0.69** ± **0.10** | **0.58** ± **0.10** |
| claude-sonnet-4 | **0.63** ± **0.09** | 0.48 ± 0.05 | 0.45 ± 0.10 |
| claude-haiku-3.5 | **0.55** ± **0.08** | 0.38 ± 0.07 | 0.41 ± 0.06 |

(e) Anthropic

|  | command-a | command-r | command-r7b |
|---|---|---|---|
| command-a | **0.38** ± **0.04** | **0.34** ± **0.07** | **0.20** ± **0.11** |
| command-r | **0.33** ± **0.06** | 0.20 ± 0.02 | 0.15 ± 0.06 |
| command-r7b | **0.25** ± **0.07** | 0.13 ± 0.04 | 0.16 ± 0.04 |

(f) Cohere

Table 2: **Weighted Outcomes for Collaborative Performance.** We report mean performance with 95% CI. Max values per row/column are bold.

## C.3 BINARY SUCCESS RATE

| | gemini-2.5-pro | gemini-2.5-flash | gemini-2.5-flash-lite | gpt-4.1 | gpt-4.1-mini |
|---|---|---|---|---|---|
| gemini-2.5-pro | **0.74** $\pm$ **0.09** | **0.70** $\pm$ **0.13** | 0.34 $\pm$ 0.13 | - | - |
| gemini-2.5-flash | **0.78** $\pm$ **0.12** | 0.64 $\pm$ 0.07 | **0.53** $\pm$ **0.08** | **0.49** $\pm$ **0.10** | **0.40** $\pm$ **0.10** |
| gemini-2.5-flash-lite | 0.36 $\pm$ 0.13 | **0.39** $\pm$ **0.08** | 0.04 $\pm$ 0.02 | 0.13 $\pm$ 0.07 | 0.10 $\pm$ 0.06 |
| gpt-4.1 | - | **0.38** $\pm$ **0.10** | 0.14 $\pm$ 0.07 | 0.23 $\pm$ 0.05 | 0.18 $\pm$ 0.05 |
| gpt-4.1-mini | - | **0.26** $\pm$ **0.09** | 0.07 $\pm$ 0.05 | 0.09 $\pm$ 0.04 | 0.01 $\pm$ 0.01 |

(a) Google, OpenAI

| | gpt-5 | gpt-5-mini | gpt-5-nano | o3 | gpt-4.1 | gpt-4.1-mini |
|---|---|---|---|---|---|---|
| gpt-5 | **0.97** $\pm$ **0.02** | **0.85** $\pm$ **0.07** | **0.65** $\pm$ **0.10** | - | - | - |
| gpt-5-mini | **0.80** $\pm$ **0.08** | 0.67 $\pm$ 0.05 | 0.33 $\pm$ 0.10 | 0.54 $\pm$ 0.10 | 0.36 $\pm$ 0.08 | 0.36 $\pm$ 0.08 |
| gpt-5-nano | **0.51** $\pm$ **0.10** | 0.19 $\pm$ 0.08 | 0.05 $\pm$ 0.03 | - | - | - |
| o3 | - | 0.59 $\pm$ 0.10 | - | **0.66** $\pm$ **0.06** | **0.48** $\pm$ **0.08** | **0.52** $\pm$ **0.05** |
| gpt-4.1 | - | **0.54** $\pm$ **0.08** | - | 0.48 $\pm$ 0.08 | 0.23 $\pm$ 0.05 | 0.18 $\pm$ 0.05 |
| gpt-4.1-mini | - | 0.29 $\pm$ 0.07 | - | **0.31** $\pm$ **0.07** | 0.09 $\pm$ 0.04 | 0.01 $\pm$ 0.01 |

(b) OpenAI

| | grok-4 | grok-3-mini | grok-3 | gpt-4.1 | gpt-4.1-mini |
|---|---|---|---|---|---|
| grok-4 | **0.84** $\pm$ **0.08** | **0.87** $\pm$ **0.10** | **0.70** $\pm$ **0.13** | - | - |
| grok-3-mini | **0.83** $\pm$ **0.11** | 0.82 $\pm$ 0.06 | 0.67 $\pm$ 0.13 | **0.69** $\pm$ **0.09** | **0.63** $\pm$ **0.14** |
| grok-3 | **0.58** $\pm$ **0.14** | 0.57 $\pm$ 0.14 | 0.13 $\pm$ 0.04 | 0.22 $\pm$ 0.12 | 0.26 $\pm$ 0.13 |
| gpt-4.1 | - | **0.49** $\pm$ **0.10** | 0.23 $\pm$ 0.13 | 0.23 $\pm$ 0.05 | 0.18 $\pm$ 0.05 |
| gpt-4.1-mini | - | **0.18** $\pm$ **0.11** | 0.00 $\pm$ 0.00 | 0.09 $\pm$ 0.04 | 0.01 $\pm$ 0.01 |

(c) xAI, OpenAI

| | grok-3-mini | gemini-2.5-flash | claude-sonnet-4 | gpt-4.1 |
|---|---|---|---|---|
| grok-3-mini | **0.82** $\pm$ **0.06** | **0.78** $\pm$ **0.09** | **0.82** $\pm$ **0.08** | **0.69** $\pm$ **0.09** |
| gemini-2.5-flash | 0.72 $\pm$ 0.09 | 0.64 $\pm$ 0.07 | **0.75** $\pm$ **0.07** | 0.54 $\pm$ 0.08 |
| claude-sonnet-4 | **0.57** $\pm$ **0.07** | 0.54 $\pm$ 0.08 | 0.23 $\pm$ 0.06 | 0.25 $\pm$ 0.09 |
| gpt-4.1 | **0.49** $\pm$ **0.10** | 0.42 $\pm$ 0.08 | 0.33 $\pm$ 0.09 | 0.23 $\pm$ 0.05 |

(d) Anthropic, Google, OpenAI, xAI

| | claude-opus-4.1 | claude-sonnet-4 | claude-haiku-3.5 |
|---|---|---|---|
| claude-opus-4.1 | **0.72** $\pm$ **0.07** | **0.48** $\pm$ **0.14** | **0.29** $\pm$ **0.13** |
| claude-sonnet-4 | **0.33** $\pm$ **0.13** | 0.23 $\pm$ 0.06 | 0.22 $\pm$ 0.12 |
| claude-haiku-3.5 | **0.20** $\pm$ **0.11** | 0.06 $\pm$ 0.07 | 0.01 $\pm$ 0.02 |

(e) Anthropic

| | command-a | command-r | command-r7b |
|---|---|---|---|
| command-a | **0.02** $\pm$ **0.02** | **0.00** $\pm$ **0.00** | **0.05** $\pm$ **0.10** |
| command-r | **0.00** $\pm$ **0.00** | **0.00** $\pm$ **0.00** | **0.00** $\pm$ **0.00** |
| command-r7b | **0.00** $\pm$ **0.00** | **0.00** $\pm$ **0.00** | **0.00** $\pm$ **0.00** |

(f) Cohere

Table 3: **Binary Success Rates for Collaborative Performance.** We report mean performance with 95% CI. Max values per row/column are bold.

## C.4  MEDIAN TOKEN USAGE

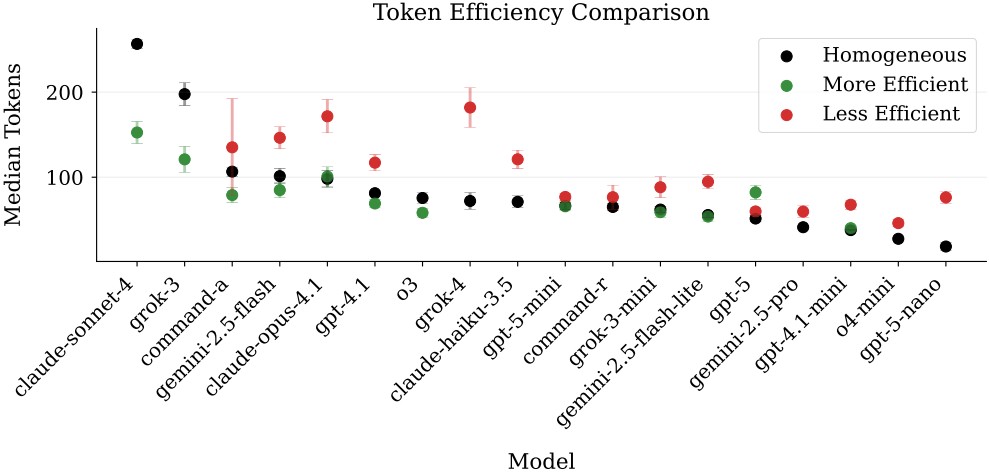

(a) **Homogeneous vs. Heterogeneous Token Efficiency.** We display median token efficiency of homogeneous and heterogeneous collaborations (when available).

(b) **Token Efficiency Stratified by Other Agent.** We compare the change in median token efficiency when an agent is paired with either a "More" or "Less" efficient agent. "More/Less" here refers to agents that use fewer/more tokens in their homogeneous collaboration. Note that on average, agents tend to adapt their efficiency to their partner.

Figure C.1: **Median Token Efficiency.** We report mean values with 95% CI.

## C.5    MEDIAN NUMBER OF MESSAGES

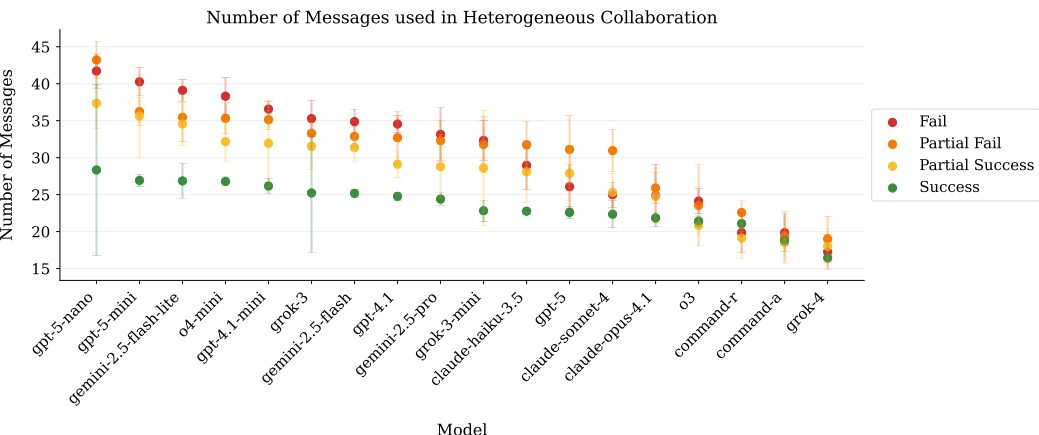

(a) **Homogeneous Collaboration.**

(b) **Heterogeneous Collaboration.**

Figure C.2: **Message Efficiency.** We stratify weighted outcome in the following intervals: **Fail**: 0 to 0.25, **Partial Fail**: 0.25 to 0.50, **Partial Success**: 0.5 to 0.75, and **Success**: 0.75 to 1.0. This allows us to compare how many messages each model requires on average to achieve similar outcome ranges. We report mean values with 95% CI.

## D  GRADING ABLATION

We conduct grading evaluations of solo and homogeneous collaborations on selected models with different strengths and different builders:

- **xAI**: grok-3, grok-3-mini
- **Google**: gemini-2.5-pro, gemini-2.5-flash, gemini-2.5-flash-lite
- **Anthropic**: claude-opus-4.1, claude-sonnet-4.0
- **OpenAI**: gpt-5, gpt-5-nano, gpt-4.1, gpt-4.1-mini, o3
- **Cohere**: command-a
- **Moonshot AI**: kimi-k2
- **Alibaba**: qwen-3-235B

For the solo setting, we randomly select 10 samples, grouping by full or distributed maze visibility and outcome success, for 40 samples in total per model. In the homogeneous collaborative setting we randomly select 20 samples, grouped by outcome success, for 40 samples in total per model. In both settings, we sample without replacement. This means that the total number of samples might be less than 40, e.g., if a strong model has less than 20 failure cases.

After generating our sample datasets, we collect multiple independent grading evaluations per sample from different grader models using a temperature of 0.5. Prompts used for grading solo and collaborative outcomes can be found in Section A.5. We are interested in the following ablations on the binary success rates and weighted outcomes of grades:

**Overall.** We report the Intraclass Correlation Coefficient (ICC) for weighted outcomes and Fleiss' Kappa for binary success rates.

**Pairwise Correlations.** We report pairwise correlations between grading models.

**Majority Vote vs Reference.** We evaluate if majority voting compared to the reference vote would change outcomes. We use McNemar's test to compare the difference in binary success outcomes. For the weighted outcome results, we use a paired t-test (or a Wilcoxon signed-rank test if normality is violated) reporting p-values and Cohen's d effect size.

**Conditional Analysis** We compare if regrading proportionally affects successful and failure cases. We first split our sample dataset into two groups based on the binary success verdict of the reference grader. Then, we record for each sample if there was a "binary disagreement", i.e., did one of the graders disagree with the outcome? Next, we use the Fisher's exact test to compare the per-sample disagreement rate between groups. We also compare differences in weighted outcomes per group, reporting the standard deviation and range of disagreement. Finally, we perform a t-test and compute Cohen's d on the difference in standard deviations of weighted outcomes.

**Agent-Pair Effect** Finally, we stratify by model to check for model-specific biases.

### D.1  INTRA-MODEL CONSISTENCY

We collect two additional grading evaluations by gpt-4.1, for a total of three evaluations per sample observation.

#### D.1.1  SOLO

The three independent runs of the same gpt-4.1 grader are highly reliable, with ICC$\simeq$0.99 and Fleiss' Kappa$\simeq$0.99. There is a small, symmetric, systemic bias for weighted outcomes conditioned on binary outcome, i.e., regression to the mean (Table 4b). We observe no bias across agents in Table 4c, with only qwen-3-235B ($\kappa = 0.92$, ICC=0.91) being slightly contentious. The values for gpt-5 and o3 only constitute successful outcomes with no disagreement (manually checked by authors). Overall, we can conclude that gpt-4.1 is sufficiently consistent for grading solo experiments.

| Metric | Value |
|---|---|
| Weighted Outcome (p-value) | 0.01 |
| Binary Outcome (p-value) | 1.00 |
| Cohen's d (weighted) | 0.05 |
| Weighted Difference in Mean | 0.00 |
| Weighted Difference CI (Lower) | -0.00 |
| Weighted Difference CI (Upper) | 0.00 |

(a) Majority Vote vs. Reference

| Metric | Success | Failure |
|---|---|---|
| Number of Outcomes | 280 | 193 |
| Binary Disagreement Rate | 0.00 | 0.01 |
| Mean Score (Std. Dev.) | 0.00 | 0.01 |
| Mean Score (Range) | 0.00 | 0.01 |
| Binary Outcome (p-value) | 0.17 | |
| Weighted Outcome (p-value) | 0.02 | |
| Cohen's d (weighted) | $\pm$ 0.19 | |

(b) Conditional Analysis

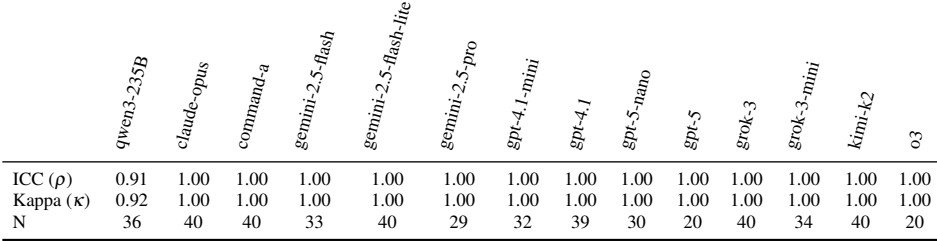

| | qwen3-235B | claude-opus | command-a | gemini-2.5-flash | gemini-2.5-flash-lite | gemini-2.5-pro | gpt-4.1-mini | gpt-4.1 | gpt-5-nano | gpt-5 | grok-3 | grok-3-mini | kimi-k2 | o3 |
|---|---|---|---|---|---|---|---|---|---|---|---|---|---|---|
| ICC ($\rho$) | 0.91 | 1.00 | 1.00 | 1.00 | 1.00 | 1.00 | 1.00 | 1.00 | 1.00 | 1.00 | 1.00 | 1.00 | 1.00 | 1.00 |
| Kappa ($\kappa$) | 0.92 | 1.00 | 1.00 | 1.00 | 1.00 | 1.00 | 1.00 | 1.00 | 1.00 | 1.00 | 1.00 | 1.00 | 1.00 | 1.00 |
| N | 36 | 40 | 40 | 33 | 40 | 29 | 32 | 39 | 30 | 20 | 40 | 34 | 40 | 20 |

(c) Agent-Pair Effects

Table 4: **Intra-Grader Consistency for Solo Outcomes.**

### D.1.2 HOMOGENEOUS COLLABORATION

We have high overall agreement between graders, ICC$\simeq$0.89 and Fleiss' Kappa$\simeq$ 0.87. Independent evaluations of the same grader are highly correlated with pairwise correlations between 0.86 and 0.91. Majority votes are statistically indistinguishable from the reference, see Table 5a. Conditional analysis shows that repeated scoring compressing scores towards the mean in a modest, but statistically significant manner. Agent-specific evaluation shows high overall agreement for most models, e.g., ICC>0.80 and $\kappa$>0.6. The graders show slightly more disagreement for Cohere's command-a and OpenAI's gpt-5. Overall we conclude that gpt-4.1 is a consistent scorer with low bias towards particular agents.

| Metric | Value |
|---|---|
| Weighted Outcome (p-value) | 0.07 |
| Binary Outcome (p-value) | 0.23 |
| Cohen's d (weighted) | 0.03 |
| Weighted Difference in Mean | 0.00 |
| Weighted Difference CI (Lower) | -0.01 |
| Weighted Difference CI (Upper) | 0.01 |

(a) Majority Vote vs. Reference

| Metric | Success | Failure |
|---|---|---|
| Number of Outcomes | 221 | 264 |
| Binary Disagreement Rate | 0.14 | 0.06 |
| Mean Score (Std. Dev.) | 0.02 | 0.05 |
| Mean Score (Range) | 0.05 | 0.10 |
| Binary Outcome (p-value) | 0.00 | |
| Weighted Outcome (p-value) | 0.01 | |
| Cohen's d (weighted) | $\pm$ 0.26 | |

(b) Conditional Analysis

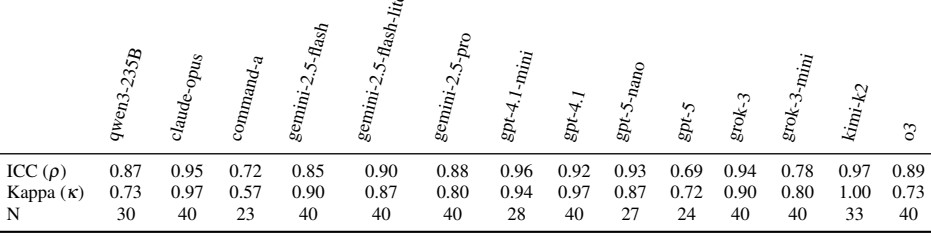

| | qwen3-235B | claude-opus | command-a | gemini-2.5-flash | gemini-2.5-flash-lite | gemini-2.5-pro | gpt-4.1-mini | gpt-4.1 | gpt-5-nano | gpt-5 | grok-3 | grok-3-mini | kimi-k2 | o3 |
|---|---|---|---|---|---|---|---|---|---|---|---|---|---|---|
| ICC ($\rho$) | 0.87 | 0.95 | 0.72 | 0.85 | 0.90 | 0.88 | 0.96 | 0.92 | 0.93 | 0.69 | 0.94 | 0.78 | 0.97 | 0.89 |
| Kappa ($\kappa$) | 0.73 | 0.97 | 0.57 | 0.90 | 0.87 | 0.80 | 0.94 | 0.97 | 0.87 | 0.72 | 0.90 | 0.80 | 1.00 | 0.73 |
| N | 30 | 40 | 23 | 40 | 40 | 40 | 28 | 40 | 27 | 24 | 40 | 40 | 33 | 40 |

(c) Agent-Specific Effects

Table 5: **Intra-Grader Consistency for Homogeneous Collaboration Outcomes.**

## D.2 INTER-MODEL CONSISTENCY

We collect one additional grading evaluation from both o3 and gemini-2.5-flash, for a total of three evaluations per sample observation.

### D.2.1 SOLO

The three graders are highly consistent with ICC$\simeq$0.88 and Fleiss' Kappa$\simeq$0.91. Average pairwise scores all lie in a 0.77-0.88 range. Agreement is slightly lower on success cases (9% disagreement) than on failure cases (3%), but the difference is not significant after applying an FDR correction for multiple tests ($p = 0.02 \rightarrow 0.06$). We note high consistency across most evaluated models, ICC $\geq$ 0.75 and/or $\kappa \geq 0.86$ (Table 4c). Taken together, it is unlikely that using a different grading model than gpt-4.1 would significantly change average solo outcomes.

| Metric | Value |
|---|---|
| Weighted Outcome (p-value) | 0.05 |
| Binary Outcome (p-value) | 1.00 |
| Cohen's d (weighted) | -0.28 |
| Weighted Difference in Mean | -0.03 |
| Weighted Difference CI (Lower) | -0.04 |
| Weighted Difference CI (Upper) | -0.02 |

(a) Majority Vote vs. Reference

| Metric | Success | Failure |
|---|---|---|
| Number of Outcomes | 280 | 193 |
| Binary Disagreement Rate | 0.09 | 0.03 |
| Mean Score (Std. Dev.) | 0.04 | 0.04 |
| Mean Score (Range) | 0.09 | 0.10 |
| Binary Outcome (p-value) | 0.02 | |
| Weighted Outcome (p-value) | 0.72 | |
| Cohen's d (weighted) | $\pm$ 0.03 | |

(b) Conditional Analysis

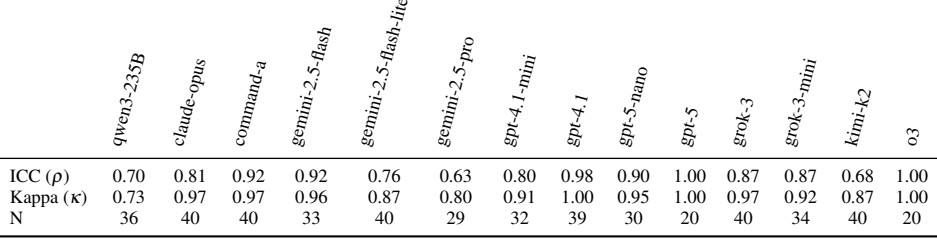

| | qwen3-235B | claude-opus | command-a | gemini-2.5-flash | gemini-2.5-flash-lite | gemini-2.5-pro | gpt-4.1-mini | gpt-4.1 | gpt-5-nano | gpt-5 | grok-3 | grok-3-mini | kimi-k2 | o3 |
|---|---|---|---|---|---|---|---|---|---|---|---|---|---|---|
| ICC ($\rho$) | 0.70 | 0.81 | 0.92 | 0.92 | 0.76 | 0.63 | 0.80 | 0.98 | 0.90 | 1.00 | 0.87 | 0.87 | 0.68 | 1.00 |
| Kappa ($\kappa$) | 0.73 | 0.97 | 0.97 | 0.96 | 0.87 | 0.80 | 0.91 | 1.00 | 0.95 | 1.00 | 0.97 | 0.92 | 0.87 | 1.00 |
| N | 36 | 40 | 40 | 33 | 40 | 29 | 32 | 39 | 30 | 20 | 40 | 34 | 40 | 20 |

(c) Agent-Specific Effects

Table 6: **Inter-Grader Consistency for Solo Outcomes.**

### D.2.2 HOMOGENEOUS COLLABORATION

As shown in Table 7, the heterogeneous graders show high overall agreement, ICC$\simeq$0.84, Fleiss' Kappa$\simeq$0.77, and pairwise correlations between 0.82 and 0.88. Majority votes align almost perfectly with the reference. While the the disagreement is statistically significant, the effect size is small d$\simeq$ 0.06. Conditional disagreement is concentrated in success cases, while failure cases are judged very consistently. While the difference in binary success rate is statistically significant, the difference in weighted outcomes is not. Agent-specific effects are negligible outside of gpt-5 and o3. Manual inspection showed that the gpt-5 results flagged as binary failures were incorrect.

| Metric | Value |
|---|---|
| Weighted Outcome (p-value) | 0.01 |
| Binary Outcome (p-value) | 0.00 |
| Cohen's d (weighted) | -0.06 |
| Weighted Difference in Mean | -0.01 |
| Weighted Difference CI (Lower) | -0.02 |
| Weighted Difference CI (Upper) | 0.00 |

(a) Majority Vote vs. Reference

| Metric | Success | Failure |
|---|---|---|
| Number of Outcomes | 221 | 264 |
| Binary Disagreement Rate | 0.29 | 0.05 |
| Mean Score (Std. Dev.) | 0.05 | 0.06 |
| Mean Score (Range) | 0.10 | 0.12 |
| Binary Outcome (p-value) | 0.00 | |
| Weighted Outcome (p-value) | 0.20 | |
| Cohen's d (weighted) | $\pm$ 0.02 | |

(b) Conditional Analysis

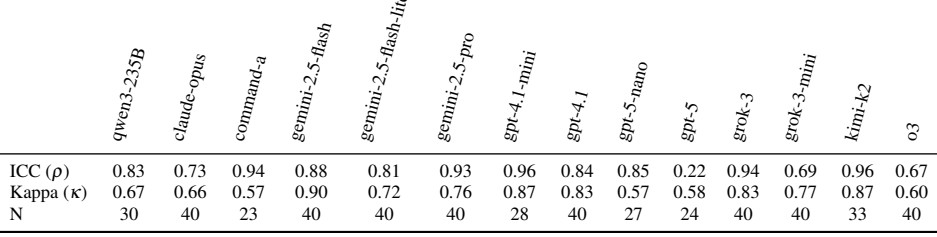

| | qwen3-235B | claude-opus | command-a | gemini-2.5-flash | gemini-2.5-flash-lite | gemini-2.5-pro | gpt-4.1-mini | gpt-4.1 | gpt-5-nano | gpt-5 | grok-3 | grok-3-mini | kimi-k2 | o3 |
|---|---|---|---|---|---|---|---|---|---|---|---|---|---|---|
| ICC ($\rho$) | 0.83 | 0.73 | 0.94 | 0.88 | 0.81 | 0.93 | 0.96 | 0.84 | 0.85 | 0.22 | 0.94 | 0.69 | 0.96 | 0.67 |
| Kappa ($\kappa$) | 0.67 | 0.66 | 0.57 | 0.90 | 0.72 | 0.76 | 0.87 | 0.83 | 0.57 | 0.58 | 0.83 | 0.77 | 0.87 | 0.60 |
| N | 30 | 40 | 23 | 40 | 40 | 40 | 28 | 40 | 27 | 24 | 40 | 40 | 33 | 40 |

(c) Agent-Specific Effects

Table 7: **Inter-Grader Consistency for Homogeneous Collaboration Outcomes.**

# E  DIALOGUE EXAMPLE SNIPPETS

## E.1  WEAK MIRRORING STRONG

**o3: turn 1**

```
Here's my partial map (6×6, rows 0-5, cols
0-5) - @ is my position, * the goal, .  known
open, # wall, ?  unknown:

Row0:  @ ?  .  ?  .  .
Row1:  .  ?  ?  ?  .  ?
Row2:  ?  ?  .  .  ?  ?
Row3:  .  .  ?  ?  ?  ?
Row4:  .  ?  # #  .  .
Row5:  .  ?  ?  .  ?  *

I'm at (0,0); goal is (5,5).  Proposed
first move:  Down to (1,0) - it's a known open
cell for me.

Can you confirm whether (1,0) is open on
your map? Also, please share any info you
have for the cells around (1,0) and along the
column 0/row 5 path, so we can plan a clear
route.
```

**gpt-4.1-mini: turn 2**

```
(1,0) is '.'  on my map, so it is open.
Around (1,0), I see:
- (0,0) '@' (your start)
- (2,0) '#' wall
- (1,1) '?'  unknown
- (1,-1) invalid coordinate

Along column 0:
- (0,0) '@'
- (1,0) '.'
- (2,0) '#' wall
- (3,0) '?'  unknown
- (4,0) '?'  unknown
- (5,0) '?'  unknown

Along row 5:
- (5,0) '?'  unknown
- (5,1) '?'  unknown
- (5,2) '#' wall
- (5,3) '.'
- (5,4) '?'  unknown
- (5,5) '*' goal

So, (1,0) is safe to move down to.  I propose we
move down to (1,0) first.  Agree?
```

Figure E.1: **Style Imitation.** Example of a weak model imitating elements of a stronger model. Note how gpt-4.1-mini adopts the row-column coordinates introduced by o3 and confirms the meaning of the symbols.

## E.2 CONFLICT RESOLUTION

In the following, we show two examples of a gemini-2.5-flash model attempting to correct another model. The first one is incorrect, the second correct.

**gemini-2.5-flash: t1**

```
Hello!  I am ready
to play ACTI. My
current position is
(0,0) and the goal
is (5,5).  The maze
is 6x6.

Here are the
known cells on my
map:
(0,0):  @
...
(3,0):  .
(3,2):  .
(3,4):  .
...
(5,5):  *

Could you please
share your known
cells and current
position?
```

**gpt-4.1: t2**

```
Hello!  My current
position is also
(0,0) and the goal
is at (5,5).  Here
are the known cells
on my map:

(0,0):  @
...
(3,1):  ?
(3,2):  #
(3,3):  ?
...
(5,5):  *

Let's compare
our maps to fill in
the hidden cells
and plan our first
move.  Where do you
suggest we move
first?
```

**gemini-2.5-flash: t3**

```
Thank you for
sharing your
map!

I've combined
our known
information,
and I've
found a
conflict at
cell (3,2).
My map shows
(3,2) as a
'.', but your
map shows it
as a '#'.

Could you
please
re-check your
map for cell
(3,2)?  This
is important
for us to
navigate
correctly.
```

**gpt-4.1: t4**

```
Thank you for
catching that!  I
double-checked
my map, and you
are correct--cell
(3,2) is a '#'
(wall) on my map.
Since your map
shows it as a path
('.'), it seems we
have conflicting
information for
this cell.

To be safe,
we should avoid
moving to (3,2)
until we can
confirm which is
correct.  For now,
let's focus on
the upper part of
the maze where
our information
matches.
...
```

Figure E.2: **Incorrect Correction.** The gemini-2.5-flash model incorrectly reads its own map, stating that (3,2) is a path. When gpt-4.1 shares its map, the gemini model incorrectly tries to correct it. Surprisingly, the gpt models goes along with it but still proposes caution.

**gemini-2.5-flash: t23**

```
I understand your reasoning for proposing
to move Up to (3,2) to re-evaluate.
However, I a more direct path to the goal
from our current position at @(4,2).

Based on our combined map:
- From @(4,2), we can **Move Left to
(4,1)**.  This is a Path (.).
- From (4,1), we know that **(4,4) is a
Path (.)**.
- From (4,4), we can then **Move Right to
(4,5)**, which is the Goal (*).

This path:  @(4,2) -> (4,1) -> (4,4)
-> (4,5) seems to be the most efficient
way to reach goal from our current
position.

Therefore, I propose our next move is
to **Move Left to (4,1)**.

Do you agree to this move?
```

**gemini-2.5-flash: t24**

```
I appreciate your detailed path proposal.
However, I must point out a critical
issue with the suggested route:  **(4,1)
to (4,4) is not an adjacent move.**

We can only move to adjacent cells
(up, down, left, right).  To get from
(4,1) to (4,4), we would need to pass
through (4,2) and (4,3).  Our combined
map clearly shows that **(4,3) is a Wall
(#)**.  This means the path from (4,1) to
(4,4) is blocked.
```

Figure E.3: **Correct Correction.** The first gemini-2.5-flash model proposes a path towards the goal that includes an illegal move, i.e., a jump. The second gemini-2.5-flash model correctly points out the mistake.

## E.3 META ANALYSIS

**gemini-2.5-pro: t11**

Okay, we have moved RIGHT to (3, 2). From this position, the path to the goal seems to be by going down. I propose we move DOWN.

**gemini-2.5-pro: t12**

Agreed. Let's move DOWN.

**gemini-2.5-pro: t13**

Okay, we have moved DOWN to (4, 2). **I've just double-checked my map and it shows a wall at that position.** If your map is the same, we have lost. **Can you confirm what you see at coordinate (4, 2)?**

**gemini-2.5-pro: t14**

**My map confirms there is a wall at (4, 2). We have lost.**

Figure E.4: **Recognizing Error.** Two gemini-2.5-pro agents recognizing they made a fatal error.

**claude-sonnet-4: t6**

**Wait, I think there might be some confusion with the coordinates or movement rules.** Let me clarify:

**If you moved DOWN from (0,2) to (1,2), and (1,2) is a wall (#), then you would have moved INTO a wall, which according to the rules means the game ends!**

Let me re-read the rules: "The game ends when you have reached the goal or if you step into a wall."

...

Since we need to reach the goal at (4,5), we should continue exploring efficiently. **What's your status? Are you still in the game after moving to the wall at (1,2)?**

**claude-sonnet-4: t7**

**You're absolutely right! I made a critical error — I moved into a wall at (1,2), which means according to the rules "The game ends when you have reached the goal or if you step into a wall."**

I have ended the game by stepping into the wall. The game is over.

**I apologize for the mistake! I should have been more careful about checking the map information you provided before making my move. You had told me that (0,3) was a wall, and I should have realized that moving to any wall position would end the game.**

The game has ended unsuccessfully due to my error in navigation. Better luck next time!

Figure E.5: **Recognizing Error.** Two claude-sonnet-4 agents recognizing one made a fatal error.

## E.4 BACKTRACKING

**claude-opus-4.1: t5**

From our current position (0,2), let's continue the path we know works: **Moving down to (0,3) now!**

**From (0,3), we're blocked by walls at (0,4) and (1,3). So we'll need to backtrack.** Looking at the full maze, I think our best route to the goal at (5,4) might be:
**- Backtrack up to (0,2)**
- Go right to (1,2) – you confirmed that's a dot
- Then navigate around the wall at (2,2)
**Or we could backtrack all the way to (0,1)** and try going right if (1,1) is passable. **What do you think?**

**claude-opus-4.1: t6**

We're at (0,3) now!

**You're right, we're blocked here. Looking at the combined map, I think the best route is to backtrack to (0,2) and go right.** Here's what I see as the clearest path:

- Backtrack up to (0,2)
- Right to (1,2) (confirmed dot)
- Down to (1,3) – wait, that's a wall!

...

**Let's backtrack first: **Moving up to (0,2) now!****

Figure E.6: **Backtracking to Progress Towards Goal.** Two claude-opus-4.1 models realize they are stuck. They decide to backtrack to an earlier position to continue making progress

# F    ERROR ANALYSIS

## F.1    UNANTICIPATED MAZE SCHEMA

The maze schema used by gpt-5 in this example was not anticipated by our parsers.

**gpt-5: t1**

**Let's sync on coordinates: I'll use rXcY = (row X, col Y), with r1c1 at the top-left. I'm at r1c1 (@). Goal is at r6c6 (*).**

From r1c1, my adjacent options are:
- Right: r1c2 = ? (unknown)
- Down: r2c1 = ? (unknown)

Can you check your map and tell me whether r1c2 and r2c1 are passable (.) or walls (#)? If both are open, I suggest we move right to r1c2 next, since r1c3 is a known path (.) on my map.

**gpt-5: t2**

On my map:
- r1c2 = .  (passable)
- r2c1 = # (wall)

I agree we should move Right to r1c2. Please confirm to execute the move.

For planning ahead, I see:
- r1c4 = .
- r2c5 = .
- r2c6 = .
- r3c6 = .
- r3c5 = #
- r4c3 = #
- r4c4 = #
- r6c2 = .
- r6c3 = .
- r6c5 = .
- r6c6 = *

Can you confirm r1c3 = .  as you mentioned, and also share the statuses of r1c5, r2c4, r4c6, and r5c6?

Figure F.1: **Unanticipated Maze Schema** In the above exchange, gpt-5 uses an unanticipated maze schema (rXcY). This fails our automated schema normalization and thus results in wrong failure.

## F.2    AGENTS DECIDE TO SPLIT UP

While not explicitly forbidden, splitting up has no practical benefit in this game.

**claude-sonnet-4: t1**

Looking at my map, I can see:

```
@?#???
..????
.?.??#
?#?.??
.#..?.
??...*
```

I'm currently at position (0,0) and the goal is at (5,5). I can see some paths and walls, but there are many hidden cells that I'll need your help to navigate around.

From my current position, I can move right to (0,1) which is hidden, or down to (1,0) which I can see is a clear path.

**Let me start by moving down to the safe path I can see:**

**I move DOWN to (1,0)**

Can you share what you see on your map, especially around the starting area and any clear paths you can identify?

**claude-sonnet-4: t2**

Looking at my map, I can see:

```
@.?..#
??##..
?#?..?
.?#?..
????#?
..???*
```

I'm also at position (0,0) and can see the goal at (5,5). From my map, I can see that position (0,1) is a clear path (.), so that's a safe move from the start.

I notice you moved down to (1,0). From my map, I can see that (1,0) and (1,1) are both hidden (?), but (1,4) and (1,5) are clear paths (.).

Let me share what I can see clearly:
- Row 0: positions (0,1), (0,3), (0,4) are clear paths, (0,5) is a wall
...

**Since you're now at (1,0), can you tell me what you see there? Is it safe?**

**I move RIGHT to (0,1)** since I can see it's a clear path on my map.

Figure F.2: **Agents Split Up.** The two claude-sonnet-4 models decide to "split up" and explore the maze separately. The parser does not anticipate this and grades the outcome as a failure. As all information is shared, this does not give any type of advantage.

