# OpenReview forum: "The Collaboration Gap"
_ICLR.cc/2026/Conference — Submitted to ICLR 2026_

### Official Review · Reviewer_gfd8 · 2025-10-20

**Soundness:** 3
**Presentation:** 3
**Contribution:** 2
**Rating:** 4
**Confidence:** 3

**Summary:**

This paper investigates the collaboration ability of multiple LLMs on tasks with communication challenges. The authors develop a maze environment. Two LLM agents involve in the system and each of them can only observe part of the maze information. They conduct experiments across multiple choices of open/closed-source LLM models, and suggests that a collaboration gap exists. The authors further provides more additional experiments on cases when two LLMs come from different model family and the relay inference setup.

**Strengths:**

Multi-agent collaboration is indeed a more and more important topics. This paper contributes to identifying key potential issues occurred in LLM cooperation tasks by investigating the LLM performance on a maza task. The paper writing is easy to follow, and the results are explained clearly. The results indeed demonstrates the existence of collaboration gap. I found some experiment results are interesting (e.g. the relay inference part).

**Weaknesses:**

1. My main concern is that the contribution in this paper, although insightful, may not reach the threshold of the acceptable of this top-tier conference. The main contribution is limited in identifying the collaboration gap, but the authors did not make progress beyond that. I believe some contribution on algorithm design to close the collaboration gap would be helpful and makes the paper stronger.

2. The experiment in this paper is limited in the maza task. It is enough to suggest "the collaboration gap" indeed exists but it may not be sufficient to suggest that such an issue appears widely in more practical domains where cooperation between LLMs are required. For example, a more practical task is applying LLM for software engineering, and it is also a valid scenario to evaluate LLM cooperations. It would be better to provide results in a more diverse range of tasks in the paper.

**Questions:**

I think the magnitude of the gap depends on how well the prompt engineering on the system/user message is. The authors suggest that one challenge in collaboration (which leads to the gap) is that the LLMs need to exchange their messages and ground their understanding "on the same page".
At least in this maze task, I think this issue can be partially solved by providing more detailed instructions in system/user prompts, such as some unified symbols/rules for communication or a clear step-by-step guidance on how to exchange information.

I'm curious about the potential of closing the gap through careful prompt engineering. Concretely, with appropriate system/user prompts, would it be possible to eliminate this collaboration gap? If the gap can be closed by prompt engineering, would it still be reasonable to claim the gap exists?

---

> ### Author Response · Authors · 2025-11-14
>
> We thank Reviewer gfd8 for their valuable feedback and for finding our results interesting!
>
> ### **On Contribution: Identifying vs. Solving (W1)**
> We respectfully note that our paper is submitted to the datasets and benchmarks track. The primary contribution is the novel benchmark itself and the rigorous empirical identification and analysis of the "collaboration gap." This is a necessary prerequisite for the community to develop the suggested "algorithm designs." Furthermore, we *do* in fact propose a novel and practical mitigation strategy, "relay inference," which is a concrete finding that partially closes the gap for weaker models.
>
> ### **On Generalizability: Mazes vs. Practical Domains (W2)**
> We used mazes as a deliberate "lower-bound" task to isolate collaborative capabilities. In Sections 2.2 and 5, "Mazes as a Lower Bound", we argue that the skills we test (establishing a shared coordinate system, merging partial information, and resolving linguistic conflicts) are fundamental prerequisites for many complex collaborations, including software engineering. A failure in this simple, isolated task strongly suggests more significant failures will occur in more complex domains.
>
> ### **On Prompt Engineering Closing the Gap (Questions)**
> This is an excellent question that gets to the core of our motivation (See also the General Comment above). The fact that the gap exists with our open-ended, "ecologically plausible" approach is the key finding. It shows that robust collaboration is not an emergent property of solo capability. Your hypothesis that the gap could be "closed by prompt engineering" strengthens our claim. We are interested in dynamic, on-the-fly communication, which is critical for real-world applications where the exact task, and thus the required scaffolding, cannot be known a-priori (Section 1, lines 35-36). The fact that models would require such heavy scaffolding (like a unified symbol set or step-by-step guidance) to succeed shows that this "on-the-fly" capability is not emergent. Additionally, in the real world we will not have control over the output format of any agents deployed by another party.

---

### Official Review · Reviewer_mYFX · 2025-10-27

**Soundness:** 3
**Presentation:** 3
**Contribution:** 3
**Rating:** 6
**Confidence:** 5

**Summary:**

This paper proposes a collaborative maze-solving benchmark to evaluate AI-AI collaboration among heterogeneous agents with partial observability. The authors evaluated 32 open- and closed-source models in solo, homogeneous, and heterogeneous pairings. Their primary finding is the "collaboration gap": models that perform well individually often experience a substantial performance drop when required to collaborate with an identical copy of themselves. They also observed that collaborative performance is heavily influenced by which agent starts the task. To address this, the paper introduces "relay inference," a strategy where a stronger agent initiates the interaction to "prime/ground" the rollout before a weaker agent takes over, which was shown to significantly boost collaborative performance and close much of the gap. The key contributions are formally defining the collaboration gap, analyzing heterogeneous collaboration dynamics, and proposing the effective relay inference strategy.

**Strengths:**

- The analysis of homogeneous, same-family (different strengths), and cross-model heterogeneous collaboration provides cool and valuable insights into agent interaction dynamics.

- The overall scope of the evaluations conducted appears quite comprehensive, covering many models and various collaboration settings.

- The proposed collaborative maze-solving benchmark is novel, isolates collaborative capabilities, and imposes minimal output constraints, which is a strong methodological contribution.

**Weaknesses:**

It is unclear whether LLM collaboration failures in this specific maze task translate to an inability to collaborate effectively in more naturalistic use cases, such as coding tasks.

Human performance on these specific maze tasks is missing, which makes it difficult to fully substantiate claims about the LLM "collaboration gap."

The reliance on the autograder is questionable, as it may introduce systemic biases or errors compared to enforcing a deterministic output format for all models.

The complexity of the ASCII map visualization raises concerns about whether perceived "collaboration failures" are actually failures of perception that could be mitigated with better data representation (e.g., natural language or tool use).

The authors should clarify the methodology of "at least 100 rollouts" by standardizing the exact number of runs across all different model types and evaluation conditions.

The discussion section does not fully justify why mazes are a good lower bound for complex collaboration, especially since the gap might be closed with different prompting or tool-use strategies. In the discussion, the authors talk about how the gap might be wider in more complex cases, but those are cases we have data for, whereas mazes seem like an esoteric example that likely doesn’t appear in the training set that much.

The section detailing the "relay inference" strategy is conceptually confusing and requires additional clarification on the mechanism and implementation.

The authors should quantify some of the interesting qualitative representational details observed during the model dialogues.

**Questions:**

- How do the authors think the collaboration failures observed in this esoteric maze might reflect failures in more representative, naturalistic language environments, such as collaborative coding or long-horizon planning tasks?
- Can the authors conduct an ablation study where the ASCII map input is replaced with a more structured, natural language (or JSON) representation to determine if performance gains are due to improved collaboration or simply better perception/parsing of the environment?
- Why not impose a strictly defined, parseable output format for the agents' moves (e.g., a specific JSON/YAML structure) to enable deterministic grading, thereby eliminating the reliance on an LLM autograder and its inherent biases?
- Can the authors include a baseline measurement of human performance on this identical distributed maze-solving task to properly contextualize the LLM failure rates?
- How many rollouts (e.g., exactly 100 or 150) were done for all model types and evaluation conditions, rather than the ambiguous "at least 100 rollouts"?

---

> ### Author Response · Authors · 2025-11-14
> **Rebuttal: Part 1**
>
> We thank Reviewer mYFX for their exceptionally thorough and insightful review! We will address each concern raised below and kindly request the reviewer to also read the General Comment above.
>
> ### **On Generalizability (W1, W6, Q1)**:
> We agree that mazes are simple. We argue (Section 2.2 and 5, “Mazes as a Lower Bound”) that the skills they test (establishing a shared coordinate system, merging partial information, and resolving linguistic conflicts) are fundamental prerequisites for many naturalistic cases, including software engineering. A failure here is a strong predictor of failure in more complex domains.
>
> ### **On Missing Human Baseline (W2, Q4):**
> This is a great suggestion. A human baseline would provide invaluable context, much as the human collaboration studies inspired this work. However, as our paper's primary focus is on AI-AI interaction and the large-scale analysis of 32 models, we believe a full human study is beyond the scope of the current work. Nevertheless, it is a clear and important next step that we will add as future work.
>
> ### **On Deterministic Output (W3, Q3)**:
> This is a deliberate and necessary design choice (also see General Comment). Our goal is to measure emergent, “on-the-fly” communication, which is a critical capability for agents in novel, real-world situations where a-priori knowledge for prompt-engineering (like a JSON schema) is unavailable or not enforceable (unable to control another party’s agent). Forcing a deterministic format would test a different, and in our view, less foundational, skill. However, we share the concern about the autograder’s reliability and possible bias (see Sections 2.1 and 7.3), which is why we performed an extensive validation (Appendix D) showing high intra- and inter-grader reliability.
>
> ### **On Perception Failure vs. Collaboration Failure (W4, Q2):**
> This is a critical point, which we answer with our "Solo: Distributed" control (Fig 4.1) (See also the General Comment above). In this setting, a solo model receives the same ASCII maps (both of them) as those given to models in the collaborative setting (one each). The fact that the solo setting succeeds, shows they can perceive and parse the representation. The failure only occurs in the collaborative setting when communication is required. This confirms it is a collaboration failure, not a perception one. As described in Section 2.1 lines 100-104, the ambiguous representation of mazes is an important feature that makes them suitable for testing collaborative capabilities using unconstrained, natural language. We will make this point more prominent in the main text.
>
> ### **Quantify the Qualitative Representations (W8)**
> That is a great suggestion, we plan to add quantitative results of selected qualitative observations as a new Appendix to the final version.

---

> ### Author Response · Authors · 2025-11-14
> **Rebuttal: Part 2**
>
> ### **On Clarity (W5, W7, Q5):**
> **Rollouts**: You are correct; this is ambiguous, thank you for pointing this out. We used "at least 100" because for cheaper models, we ran additional rollouts to improve the statistical confidence of our results. As each rollout can take up to 50 turns, rollouts for more expensive models are costly (see Section 7.4). We will clarify this methodology and update the main text and appendix accordingly in the final version.
>
>
> **Relay Inference**: We apologize that this section was conceptually confusing. We will rewrite it for clarity, adding a more explicit step-by-step description of the "relay" mechanism both in the main text and in the appendix. The core conceptual motivation is: _"if one needs to deploy a model to collaborate on a task with another model, can a “smart” deployment strategy using models with different strength/cost profiles offer efficiency gains?_"
>
> The basic mechanism works as follows:
>
> Assume two models: a stronger but more expensive model A, and a weaker but cheaper model B. Further assume that our opposing party chooses to deploy the cheaper model (B) to save costs.
> 1.  We first generate a maze trajectory using A and B following the usual setup outlined in Section 2.1. This results in a sequence of interactions, $\tau = [a_0, b_1, a_2, b_3, …, a_{N-1}, b_N]$.
> 2.  Next, we explore the importance of the initial turns: we take the first k messages of $\tau$, e.g., for k = 2, we have $\tau(k=2) = [a_0, b_1]$. Then, instead of using A and B to continue the trajectory as before, we “relay” the trajectory using two independent instances of the cheap model, B1 and B2.
> 3.  By varying k, this results in several “relay” trajectories, where the first k interactions are produced by the pair A, B, but the remaining interactions by two independent instances of model B. For example, relay($\tau(k=2)$, B1, B2) $= [a_0, b^1_1] \to$  $[b^2_2, b^1_3, …, b^2_{N-1}, b^1_{N}]$, where the superscript indicates B1 or B2, and the subscript the turn index.
>
> This provides insights into the importance of the initial interactions to effectively “ground” the communication. Alternatively, we can explore how well a stronger model can “recover” collaborations started by deploying a weaker model. This is done by creating the initial trajectory using two independent copies of B, and instead doing a “relay” at turn k using the stronger model A. Our results show (Section 4.4) that using a strong “primer” model is significantly more effective than using a strong “recovery” model.

---

> > ### Comment · Reviewer_mYFX · 2025-11-26
> >
> > Thank you to the authors for their response to my review. I have a few additional followup questions:
> >
> > - Your point about mazes being simple is not quite entirely what my previous comment was trying to get at. While there is a performance drop between the single agent and multiple agents solving the task, it is not obvious that this gap translates to other settings such as coding. The reason being is that there is simply more examples of code being successfully written, so the agents might have a better chance of collaborating with users despite not observing their underlying intentions. In contrast, there might be very few datapoints of collaborative maze solving these models are trained on. Failure here would not indicate a universal inability to collaborate, but rather an inability to collaborate on maze tasks. Do the authors have intuitions or concrete preliminary results supporting the hypothesis that failures to collaborate on these mazes translates to universal failures on cooperation tasks?
> >
> > - What is the concrete plan for quantifying the qualitative findings?
> >
> > - Increasing number of rollouts to boost significance seems strange and prone to deceiving some readers. I understand the need to run more expensive models for less time. However, can the authors please provide reports of the mean and significance of even the cheaper models on the same smaller set of data? This would make the comparison between models more fair.

---

> > > ### Author Response · Authors · 2025-11-28
> > > **Rebuttal 2: Part 1**
> > >
> > > ### **On the Interpretation and Transferability of Results**
> > >
> > > **Interpretation**
> > > The reviewer raises an important point regarding the prevalence of specialized training data. We agree that a lack of specific “collaborative maze-solving datapoints” in the training set could explain the performance gap. However, this observation actually supports our central claim rather than refuting it.
> > >
> > > Our research specifically set out to answer if "_current training strategies create agents capable of dynamic collaboration_" (Section 1, l40-41). Phrased differently: can we expect dynamic collaboration capabilities to materialize as an emergent capability, or, does this require specialized interventions? The observed gap in collaborative maze-solving demonstrates that collaborative capabilities are not automatically acquired alongside solo mastery. Even if the remedy is simply to include more data, the necessity of such targeted intervention supports interpreting our results as an “_existence proof that effective collaboration represents a distinct axis of capability that current training strategies fail to capture_" (Section 1, l91-92), even in models that have mastered the underlying task.
> > >
> > > Note that we do not claim our results indicate a  "_universal inability to collaborate_," or that dynamic collaborative capabilities might not already be present for some tasks. Rather, we claim that high solo performance on a specific task is not a sufficient condition for collaborative performance on that same task.
> > >
> > > **Transferability**
> > > As we discuss in Section 2.2, successful collaboration on our maze task requires several core capabilities like “grounding” and conflict resolution that are strongly associated with collaborative performance in humans.
> > >
> > > - **The parallel**: We argue that our “split-maze” task is a structural proxy for partial observability that can arise in collaborative software engineering. Just as two agents in our maze must reconcile different map views, coding agents can face a “split context”, where each agent sees only a subset of the files. This might happen if a code base does not fit into an agent’s context, or if agents have different role-based access controls. Echoing the maze setup, the agents must agree on the folder structure to organize new files or import specific functionality, and align on coding conventions, package versions, shared dependencies, and resolve the occasional warnings and errors.
> > >
> > > - **Empirical evidence**: Recent work by [1] that analyzed common failures in multi-agent tasks supports this link, finding that "_failures arising from ineffective communication, poor collaboration, [and] conflicting behaviors among agents_" (Section 1, l70-71) represent an important failure mode. The underlying models in this study represent a subset of the previous generations of the frontier models studied in our work, e.g., GPT-4, Claude 3, and Qwen 2.5, tested on coding, math, and general-agent tasks. This suggests our benchmark captures collaborative capabilities whose relevance extends beyond mazes.
> > >
> > >
> > > [1] Cemri, Mert, et al. "Why do multi-agent llm systems fail?." NeurIPS (2025).
> > >
> > >
> > >
> > > ### **Quantifying Qualitative Results**
> > >
> > > We formalized our qualitative analysis into the following approach:
> > > 1. We manually inspected many hundreds of rollouts. Qualitative observations of interest were logged and gathered, then used to group several behaviors of interest, e.g., conflict resolution, proposal dominance, maze representations, global/local strategies, etc..
> > > 2. Next, we designed a prompt capturing a selection of these behaviors and used an LM to inspect sampled rollouts for uncaptured behaviors of potential interest. This resulted in additional suggestions.
> > > 3. We designed a final prompt summarizing our main qualitative features of interest and used an LM to annotate a sample of rollouts.
> > > 4. We are currently performing human-verification on the LM annotations to ensure high agreement.
> > > 5. The final version will include an Appendix reporting on the quantitative frequency of these behaviors, providing a statistical view on collaborative behaviors.

---

> > > > ### Author Response · Authors · 2025-11-28
> > > > **Rebuttal 2: Part 2**
> > > >
> > > > ### **Robustness Check**
> > > >
> > > > Per the reviewer’s request, we performed a sensitivity analysis by randomly down-sampling the higher-resource experiments to N=100 observations. We find the identified collaboration gap remains robust (see Table below). While confidence intervals widen due to reduced sample size, the mean performance remains stable. For some weaker models with high variance (gpt 4.1 nano, llama-3.3 70B), the widened CIs at N=100 occasionally overlap, reflecting the expected loss of statistical power.
> > > >
> > > > Results are reported as the mean weighted outcome with 95% confidence intervals:
> > > >
> > > > | Model | Solo Full | Solo Distributed | Collaboration Distributed |
> > > > | :--- | :--- | :--- | :--- |
> > > > | qwen-3-235B | 0.984 ± 0.019 | 0.735 ± 0.070 | 0.362 ± 0.050 |
> > > > | qwen-2.5-72B-it | 0.511 ± 0.058 | 0.549 ± 0.055 | 0.341 ± 0.049 |
> > > > | phi-4 | 0.535 ± 0.061 | 0.493 ± 0.058 | 0.353 ± 0.054 |
> > > > | o4-mini | 1.000 ± 0.000 | 0.940 ± 0.047 | 0.746 ± 0.064 |
> > > > | llama-4-scout | 0.449 ± 0.056 | 0.467 ± 0.052 | 0.274 ± 0.038 |
> > > > | llama-4-maverick | 0.576 ± 0.061 | 0.491 ± 0.062 | 0.420 ± 0.054 |
> > > > | llama-3.3-70B-it | 0.530 ± 0.054 | 0.397 ± 0.065 | 0.381 ± 0.043 |
> > > > | kimi-k2 | 0.943 ± 0.037 | 0.769 ± 0.065 | 0.476 ± 0.057 |
> > > > | gpt-oss-20b | 0.987 ± 0.019 | 0.629 ± 0.074 | 0.283 ± 0.057 |
> > > > | gpt-oss-120b | 1.000 ± 0.000 | 0.994 ± 0.012 | 0.797 ± 0.062 |
> > > > | gpt-5-nano | 1.000 ± 0.000 | 0.878 ± 0.059 | 0.345 ± 0.050 |
> > > > | gpt-5-mini | 1.000 ± 0.000 | 0.982 ± 0.017 | 0.879 ± 0.051 |
> > > > | gpt-4.1-nano | 0.398 ± 0.085 | 0.301 ± 0.067 | 0.291 ± 0.043 |
> > > > | gpt-4.1-mini | 0.993 ± 0.014 | 0.775 ± 0.064 | 0.399 ± 0.047 |
> > > > | gpt-4.1 | 0.964 ± 0.028 | 0.843 ± 0.058 | 0.546 ± 0.069 |
> > > > | gemma-3-27b-it | 0.493 ± 0.058 | 0.486 ± 0.052 | 0.330 ± 0.051 |
> > > > | gemini-2.5-flash-lite | 0.788 ± 0.067 | 0.594 ± 0.084 | 0.303 ± 0.056 |
> > > > | gemini-2.5-flash | 0.972 ± 0.032 | 0.893 ± 0.055 | 0.824 ± 0.066 |
> > > > | deepseek-v3 | 0.652 ± 0.081 | 0.576 ± 0.066 | 0.383 ± 0.054 |
> > > > | deepseek-R1 | 1.000 ± 0.000 | 1.000 ± 0.000 | 0.733 ± 0.059 |
> > > > | command-r7b | 0.312 ± 0.046 | 0.249 ± 0.047 | 0.174 ± 0.034 |
> > > > | command-r | 0.333 ± 0.059 | 0.334 ± 0.044 | 0.204 ± 0.032 |
> > > > | command-a | 0.585 ± 0.058 | 0.539 ± 0.060 | 0.360 ± 0.049 |
> > > > | claude-haiku-3.5 | 0.498 ± 0.050 | 0.217 ± 0.066 | 0.435 ± 0.054 |

---

### Official Review · Reviewer_cd33 · 2025-11-01

**Soundness:** 2
**Presentation:** 3
**Contribution:** 2
**Rating:** 2
**Confidence:** 3

**Summary:**

The authors introduce a new maze based collaborative test for RL algorithms and test it on LLMs. They show that their test correlates with model "power" and provide suggestions on how to make LLMs more collaborative.

**Strengths:**

# originality

Evaluating collaboration between AI agents has been an ongoing area of research for decades. This appears to be a variant of the overcooked test [1], but with lower complexity and partial information. As they do not discuss how this fit into the literature it is hard to evaluate the originality.


# quality

The experiments appear to have been done well, and they test a large number of LLM variants. The use of an automated LLM as grader is concerning, but they discuss the issues with it and seem to have done things correctly. The lack of code release though means I cannot verify these statments.

# clarity

The paper is clear, giving good examples and explains what is happening well. I think putting a full dialogue somewhere prominent (maybe a section in the appendix) would help as I had to dig around in the appendix to understand exactly what transpires in a run.

# significance

As mentioned above I have concerns about how this paper engages with the literature. Taking it on face value there are better and much more established tests for AI agent collaboration (Hanabi, Diplomacy, Overcooked, ...), so I'm not sure if this adds much to the disucssion

[1] Carroll, M., Shah, R., Ho, M.K., Griffiths, T., Seshia, S., Abbeel, P. and Dragan, A., 2019. On the utility of learning about humans for human-ai coordination. Advances in neural information processing systems, 32.

**Weaknesses:**

The 6x6 maze seems very small, A* can solve that trivially.

The use of a grader AI adds an additional level of complexity to the experiment.

The authors don't appear to engage with the SOTA in collaborative AI, instead focusing on LLMs only.

Most of my other concerns are in the other sections, I think if the paper significantly toned down it's claims it would be publishable.

**Questions:**

I'm concerned about the reproducibility of this experiment. Why didn't the authors include the code with the submission? This is a complex simulation, code is need for other groups to reproduce the results. Will the the authors do a full code release if the paper is published? The reproducibility statement only talks about the license not access.

I find the use of anthropomorphic language (e.g. on line 302 "The stronger o3 immediately seeks to") concerning. Are the authors arguing that LLMs have goals? I think discussing what is happening with more theoretically grounded language would improve the paper.

Hanabi[1] is the standard for collaborative text based AI research. Why is this maze approach better and how do results compare to Hanabi? More generally was AI collaboration in non-LLM settings discussed at all in the paper?

How do these models perform when paired with non-LLM partners? Training an RL agent to solve these mazes seems trivial.

Can the models share their maps to each other? Was their any filtering of messages?

[1]Bard, N., Foerster, J.N., Chandar, S., Burch, N., Lanctot, M., Song, H.F., Parisotto, E., Dumoulin, V., Moitra, S., Hughes, E. and Dunning, I., 2020. The hanabi challenge: A new frontier for ai research. Artificial Intelligence, 280, p.103216.

---

> ### Author Response · Authors · 2025-11-14
>
> We thank Reviewer cd33 for their feedback. We believe there is a core misunderstanding of our paper's motivation and contributions, which we hope to clarify.
>
> ### **On Goals and Positioning vs. Hanabi/Overcooked (Summary, Originality, Significance, W3, Q3, Q4):**
> As discussed in the general comment above, our work is not a "variant of overcooked" or a competitor to Hanabi. Those are (great!) benchmarks for MARL and game-theoretic coordination with highly constrained, symbolic action spaces. Noted in lines 65-66 of the Introduction, a key difference in agents powered by language models is the opportunity to use rich, natural language. However, this shift from communication through constrained, symbolic action spaces to natural language introduces novel collaboration challenges. This is why our work is inspired by and builds upon a different, and highly relevant, body of literature: seminal studies on human-human collaboration (Section 2.1, e.g., Garrod & Anderson, 1987). Our work is the first to adapt this methodology to AI-AI collaboration at scale, focusing on a fundamentally different and new problem: evaluating collaboration via unstructured, open-ended natural language dialogue. Consequently, this is why the scope of this research is collaboration between agents powered by language models.
>
> ### **On Task Triviality** ("A* and RL can solve mazes trivially") (W1, Q4):
> The point of our benchmark is not to measure pathfinding capabilities. Instead, the capability of interest is collaboration under partial observability using only unconstrained natural language. With that in mind, we designed a benchmark that allows us to disentangle "raw", task-specific capabilities from those needed for collaboration (Section 2.1, lines 105-107).
>
> Our "Solo: Distributed" baseline (Fig 4.1) displays this. In this control, a solo model is given both partial maps and solves the task with a high solve rate. This shows the models can parse the ASCII and can solve the pathfinding problem. The "collaboration gap" only appears when a second agent is introduced and communication is required. This shows the failure is one of collaboration and grounding, not of pathfinding.
>
> Additionally, we provide ablations of different maze sizes and wall densities in Appendix B. We chose the 6x6 maze dimension to ensure most of the 32 models studied would have a reasonable solo solve-rate.
>
> ### **On Reproducibility (W2, Q1):**
>
> Grader: We used the AI grader because we evaluate unconstrained natural language, where a deterministic parser is impossible. We share the concern about its reliability (Section 7.3), which is why we performed an extensive validation (Appendix D) showing high intra- and inter-grader reliability.
>
> Code Release: As mentioned in the reproducibility statement, line 488: "_We plan to open-source the code used to conduct the experiments described in this work._" In the meantime, we shared all prompts used in our work as well as an in-depth discussion of our implementation considerations in Appendix A.
>
> Confidence Intervals: Due to the stochastic nature of our experiments (Section 7.2), we performed a large number of rollouts for each of the evaluated models and settings, reporting 95% confidence intervals.
>
> ### **Other Questions (Clarity, Q2, Q5):**
> Dialogue Examples: We apologize for insufficient references in the main text. We included numerous snippets (e.g., Figs 2.2, 4.2, 4.4, 4.5, E.1-E.6, F.1-F.2) to highlight specific behaviors, as full transcripts (30-50 messages) were impractical for the appendix. We will add a clear pointer in the main text to the appendix sections containing these examples and will add additional (truncated) transcripts to make it easier to visualize this task.
>
> Anthropomorphic Language: This is a fair and important point, thank you for raising it. We will revise the text to remove anthropomorphic language (e.g., "seeks to") and use more precise descriptions.
>
> Filtering Messages: Messages were not filtered in any way, allowing models to share partial, full, or no map details at all. As shown in e.g., Fig 2.2, 4.4, and E.1, this can lead to a variety of behaviors.

---

> > ### Comment · Reviewer_cd33 · 2025-11-15
> >
> > I have read the authors comments and the other reviews, I maintain my rating
> >
> > Again, as I said my issue is with the strength and breadth of the claims. The abstract says "guidance that applies to AI–AI and human–AI collaboration", but you claim that the large existing body of research on this is not relevant because you use an LLM. I asked about the foundational works, but there are example in the literature that involve language (even unstructured, open-ended natural language dialogue) [1]. If you want to say the paper is looking at the limitation of LLMs-LLM collaboration and LLM-Human collaboration I wouldn't have this concern, the issue is the scope of the claims.
> >
> > Again, I mentioned code release. You say in the paper it will be _open-sourced_ , that is not a commitment to release it is a license. What are the plans for a code release? Will the code release be capable of reproducing all the experiments?
> >
> > What does MARL stand for? I don't see it defined in the message
> >
> > [1] Meta Fundamental AI Research Diplomacy Team (FAIR)†, Bakhtin, A., Brown, N., Dinan, E., Farina, G., Flaherty, C., Fried, D., Goff, A., Gray, J., Hu, H. and Jacob, A.P., 2022. Human-level play in the game of Diplomacy by combining language models with strategic reasoning. Science, 378(6624), pp.1067-1074.

---

> > > ### Author Response · Authors · 2025-11-15
> > >
> > > We thank Reviewer cd33 for the swift follow-up and for sharing their remaining concerns.
> > >
> > >
> > > We certainly do not intend to overgeneralize our contributions and apologize for possible miscommunication. Truly, this was not our intention. We will update the relevant portions (abstract, conclusion) to emphasize that the scope of our claims relate to language models (LMs), **not** all of AI. Thank you for reiterating this point, we trust that we hereby have alleviated your concern.
> > >
> > >
> > > As written in our General Comment, we also do not claim that the existing work on cooperative AI is not relevant, and are committed to extending our review of related work to better contextualize our contributions. Please allow us to use your excellent reference to the Diplomacy (Cicero) paper to make our specific contribution sharper. The goals of the two papers are fundamentally different:
> > > - **Diplomacy (Cicero)'s Goal:** The goal was to create a purpose-built, hybrid system to achieve human-level performance in one specific, complex, and known domain (Diplomacy). As the paper details, this required a massive, multi-component architecture, including:
> > >   - A strategic reasoning module trained with RL to plan actions.
> > >   - An LM fine-tuned on an enormous in-domain dataset (>40k games, >12M messages).
> > >   - Complex engineering to use the planner's "intents" to control the LM's dialogue output.
> > >   - A suite of 16+ classifiers and heuristics to filter the LM's outputs.
> > >
> > > In this light, clarifying that our work is focused on pretrained generalist LMs — thanks again for having insisted on this point! — will help further clarify the distinction from Cicero.
> > > - **Our Goal**: Our goal is to evaluate the emergent, adaptive, "on-the-fly" collaboration capabilities of general-purpose LMs. We start from the premise that for most real-world tasks, it is unrealistic to assume we will have a-priori knowledge or a large, domain-specific dataset to build a bespoke system like Cicero. We are therefore testing the "raw" collaborative capability of the LM itself, acting end-to-end as both planner and communicator without heavy, task-specific scaffolding. The "collaboration gap" is significant precisely because it reveals a failure in this raw, emergent skill, which a system like Cicero's was explicitly engineered to overcome.
> > >
> > > **On Code Release**: To avoid confusion about the term open source: **We will make the full benchmark source code and evaluation scripts publicly available upon publication**. The repository will contain all necessary components to reproduce all the experiments presented in the paper. However, as noted in Section 7.4, some of these experiments carry a considerable cost. We are strong believers in open science and hope that our research can help other researchers and practitioners work on this important direction.
> > >
> > > **On MARL**: We apologize for the jargon. The cited works (Overcooked, Hanabi, Diplomacy) are foundational for the research area generally referred to as Multi-Agent Reinforcement Learning (MARL). Our paper investigates a different research question: how pre-trained LMs handle collaboration challenges emergently, as opposed to MARL agents that are often trained with task-specific architectures and datasets.
> > >
> > >
> > > Thank you again for this discussion, it has been very helpful.

---

> > > > ### Comment · Reviewer_cd33 · 2025-11-18
> > > >
> > > > Previously you said:
> > > >
> > > > > We interpret our results as an existence proof that effective collaboration represents a distinct axis of
> > > > capability that current training strategies fail to capture. The gap is a critical roadblock on the path to
> > > > safely and effectively deploying general-purpose agents.
> > > >
> > > > Which became:
> > > >
> > > > > evaluating emergent collaboration via unstructured, open-ended natural language dialogue
> > > >
> > > > Now you say:
> > > >
> > > > > evaluate the emergent, adaptive, "on-the-fly" collaboration capabilities of general-purpose LMs
> > > >
> > > > This new framing seems like it squarely fits in the existing AI collaboration literature, measuring how agents can generalize is one of the stated goals of the Hanabi challenge. Much of the research in this area is focused on creating generalist learners (e.g. muzero, dreamerV3, etc) that adapt to novel situations, is that not what you are describing? I'm trying to understand what the contribution is, and if it is novel
> > > >
> > > > >  We apologize for the jargon
> > > >
> > > > Why is jargon incorrect to use here?

---

> > > > > ### Author Response · Authors · 2025-11-18
> > > > >
> > > > > We thank Reviewer cd33 for the continued engagement. We appreciate the opportunity to clarify the distinction between our work and the cited cooperative AI and generalist learning literature (Hanabi, MuZero, DreamerV3).
> > > > >
> > > > >
> > > > > ### **Evaluating Learning Algorithms (RL) vs. Frozen Capabilities of Pre-Trained Models (LMs)**
> > > > > > _”Much of the research in [AI collaboration] is focused on creating generalist learners (e.g. muzero, dreamerV3, etc.) that adapt to novel situations, is that not what you are describing?”_
> > > > >
> > > > > To clarify the difference, we find it helpful to compare to the human standard of collaboration: humans are surprisingly effective at collaborating “on the fly” using natural language, without pre-defined protocols or task-specific training. Our study investigates if widely deployed, pre-trained LMs share this capability.
> > > > >
> > > > > Cited benchmarks like Hanabi and Overcooked are designed to measure how well an algorithm can *learn* a policy to solve a coordination problem, often dealing with restricted symbolic action spaces. In contrast, our work targets a different problem space:
> > > > >
> > > > > - **Subject**: we are not evaluating the adaptability of *learning algorithms* (like DreamerV3, MuZero) that update weights to adapt over many episodes. We are evaluating frozen, pre-trained LMs via (zero-shot) inference.
> > > > > - **Modality**: we focus on unconstrained natural language, rather than symbolic action spaces. The generalist learners mentioned (MuZero, DreamerV3) operate on defined observation/action spaces (pixels to actions). They cannot natively participate in this benchmark because they lack the architecture to parse and generate open-ended natural language.  To apply them here would require turning them into language models, validating our focus on LMs as the relevant subject.
> > > > > - **Goal**: we seek to determine if adaptive collaborative capabilities are emergent skills in current LMs, or if they represent a distinct capability axis not covered by current LM training strategies.
> > > > >
> > > > > We thus do not propose a new learning algorithm, or a benchmark for adaptive learning algorithms; we provide an existence proof that current state-of-the-art LMs can suffer a significant degradation when forced to coordinate via natural language, despite having the raw reasoning capabilities to solve the task alone. We focus on collaboration using unconstrained natural language to simulate and anticipate efforts to directly deploy LMs to power agent-like solutions.
> > > > >
> > > > >
> > > > > ### **Contributions and Novelty**
> > > > > A key challenge our benchmark design tackles is disentangling “raw” task capabilities (e.g., perception, pathfinding) from those needed for collaboration (e.g., grounding).
> > > > >
> > > > > 1) Identifying the “Collaboration Gap”: we use our novel benchmark to provide empirical evidence that leading LMs can possess the “raw” reasoning capabilities to solve a task (as shown in our solo control), yet suffer a significant degradation when forced to collaborate via unguided natural language. The fact that the same model can solve a task alone but performs significantly worse when required to collaborate *with a copy of itself* is a non-trivial finding. For example, it highlights that only evaluating LMs for individual performance can be misleading if the goal is to deploy them in collaborative settings.
> > > > >
> > > > > 2) Dynamics of Heterogeneous Collaboration between LMs: by pairing LMs with different strengths, we found that collaborative performance can be strongly influenced by interaction order.
> > > > >
> > > > > 3) Based on (2), we propose and validate “relay inference.” This is a practical deployment strategy showing that the collaboration gap identified in (1) can be significantly reduced for weaker models by strategically priming the interaction using a stronger model, a finding specific to the inference dynamics of LMs.
> > > > >
> > > > >
> > > > > ### **Consistency of Claims and Jargon**
> > > > > We apologize if our attempts to be precise appeared as shifting claims. All three statements you quoted describe the same statement from different perspectives:
> > > > >
> > > > > This is our core finding:
> > > > > > We interpret our results as an existence proof that effective collaboration represents a distinct axis of capability that current training strategies fail to capture. The gap is a critical roadblock on the path to safely and effectively deploying general-purpose agents.
> > > > >
> > > > > This is the mechanism:
> > > > > > evaluating emergent collaboration via unstructured, open-ended natural language dialogue
> > > > >
> > > > > This is the scope of our study:
> > > > > > evaluate the emergent, adaptive, "on-the-fly" collaboration capabilities of general-purpose LMs
> > > > >
> > > > > Regarding “Jargon”: The use of jargon is not incorrect, we simply meant to apologize for using the acronym “MARL” in our previous response without defining it, which we realized might have caused confusion.
> > > > >
> > > > > We hope this clarifies the boundary between our study of frozen, pre-trained LM capabilities and the existing literature on cooperative AI and generalist learning.

---

> > > > > > ### Comment · Reviewer_cd33 · 2025-11-19
> > > > > >
> > > > > > My concern is with the novelty to the literature (and strength of contribution), not novelty of the test. It appears your main claim is that you should only be compared to offline learners but you also mention: 'emergent, adaptive, "on-the-fly" collaboration capabilities' as goals so online learns seem to be what you are comparing to. As I said before, my main concern is that looking at how the claims related to the results puts you over your skiis.
> > > > > >
> > > > > > For example you say:
> > > > > >
> > > > > > > The fact that the same model can solve a task alone but performs significantly worse when required to collaborate with a copy of itself is a non-trivial finding.
> > > > > >
> > > > > > You claim this is a non-trivial finding, but do not have any citations or discussion to back up the claim. My understanding is that collaboration imposing overhead is not a new result, so claims like this make me concerned about the entirety of the paper's novelty. Note the reviewer guidelines are:
> > > > > >
> > > > > > >  is the submission clear, technically correct, experimentally rigorous, reproducible, does it present novel findings (e.g. theoretically, algorithmically, etc.)?
> > > > > >
> > > > > > On a final note, remember that as per the guidelines:
> > > > > >
> > > > > > > authors disclose any significant usage of LLMs in research ideation/writing.

---

### Author Response · Authors · 2025-11-14
**General Comment to All Reviewers and Chairs**

We thank all three reviewers for their time and valuable feedback. We have identified two core methodological critiques that we wish to address head-on, as they are central to our paper's thesis and contributions.

### **On the Benchmark's Goal: Unconstrained Natural Language vs. Symbolic Multi-Agent Reinforcement Learning**

A key question (raised by R-cd33) is how our benchmark relates to existing AI collaboration tasks like Hanabi or Overcooked. We must clarify that our work is not a competitor to MARL benchmarks, which test coordination via constrained, symbolic action spaces (e.g., "play card"). As noted in lines 65-66 of the Introduction, a key difference in agents powered by language models is the opportunity to use rich, natural language. However, this shift from communication through constrained, symbolic action spaces to natural language introduces novel collaboration challenges. Our benchmark is thus inspired by and builds upon a different body of literature: seminal studies on human-human collaboration (Section 2.1, e.g., Garrod & Anderson, 1987). Our work adapts this methodology to AI-AI collaboration at scale, focusing on a different and novel problem: evaluating emergent collaboration via **unstructured, open-ended natural language dialogue**. We will extend our review of related work to make this distinction clearer and properly attribute past efforts.


### **On the "Gap": A Failure of Collaboration, Not Perception or Task Design**

- **Is it a Perception Failure?**
Our “Solo baselines (beginning of section 3 and Fig 4.1)" directly answer this. In this control, a solo agent is either given the full ASCII map (Full) or both partial ASCII maps (Distributed). In both cases the solo agent succeeds with a high solve-rate. This shows the studied models can parse the ASCII and can solve the pathfinding problem, even when information is distributed. The performance drop only occurs when they must communicate with a partner. This isolates the failure to collaboration, not perception. We will update Section 3 and the discussion of results shown in Sections 4.1 and 4.2 to make the purpose and outcomes of this control more apparent.


- **Can it be "Prompt-Engineered" Away?**
This question (from R-gfd8, R-mYFX) is central to our paper's motivation. We are interested in adaptive, "on-the-fly" communication. In the open, unstructured real world, it is unrealistic to assume we will always have the a-priori knowledge required to pre-define a perfect communication protocol (like a JSON schema) for every novel task an agent might encounter. Additionally, we will not have control over the output format of any agents deployed by another party. The fact that current models fail in the absence of heavy scaffolding shows that collaboration is not an emergent property of solo capabilities. Our benchmark is the first to quantify this "native" collaboration gap, which has clear implications for further integration of AI agents into society.

### **Our Contributions**
Having clarified our motivation and methodology, we wish to re-emphasize our contributions in the context of the datasets and benchmarks track. Our work provides a comprehensive empirical study (evaluating 32 leading language models) that identifies a critical, novel failure mode. The key findings include:
- The formal identification and quantification of the "collaboration gap" in unconstrained natural language communication.
- The discovery that this gap is especially severe in distilled models.
- A robust analysis of heterogeneous collaboration, revealing that "ordering effects" (who speaks first) are a critical performance determinant.
- A novel, practical mitigation strategy, "relay inference," where a strong agent "primes" the interaction, which we show closes much of the gap displayed by weaker models.

We are confident these are significant contributions and will revise the paper to make these points clearer.

---

### Meta-Review · Area_Chair_dCmo · 2026-01-08

**Summary:**

The paper seems originally motivated by the fact that while agents are capable of more and more, collaborative interactions do not seem to be improving at the same rate as everything else. They attempt to explore this via experiments in a bespoke environment. This leads to the primary concern of the reviews, whether one can draw any meaningful conclusions given the lack of established benchmarks in the field, in addition to concerns regarding a lack of human baselines, etc. There seems to be a "gap" in between the high level claims and the experimental evidence.

**Reviewer Concerns:**

The rebuttal was extensive and clarified many of the concerns but would likely require a substantial rewrite of the original paper to actually make work (reduce scope to LLM collab instead of all of AI). Other concerns such as generalization of results in mazes to other areas are also not addressed. I share some of the concerns as the reviewers despite the rebuttal comments. Overall, I would put this paper in the bucket of "potentially promising but not quite ready for publication" and would encourage the authors to make all the changes they have promised in the rebuttals before resubmitting in the future.

**Reviewer Scores:**

cd33 would not increase their score but I would also discount their review as they mention new concerns unrelated to the original ones in subsequent discussion

mYFX is also unlikely to increase their score as there appears to be some difference in the interpretation to their questions

gfd8's concerns also do not appear to be fully addressed and so would be unlikely to increase their score

---

### Decision · Program_Chairs · 2026-01-26

Reject